# Data-Driven Discovery of Dynamical Systems in Pharmacology using Large Language Models

**Samuel Holt**[*]
University of Cambridge
sih31@cam.ac.uk

**Zhaozhi Qian**[*†]
Elm UK
zqian@elm.sa

**Tennison Liu**
University of Cambridge
tl522@cam.ac.uk

**James Weatherall**
AstraZeneca

**Mihaela van der Schaar**
University of Cambridge
mv472@cam.ac.uk

## Abstract

The discovery of dynamical systems is crucial across a range of fields, including pharmacology, epidemiology, and physical sciences. *Accurate* and *interpretable* modeling of these systems is essential for understanding complex temporal processes, optimizing interventions, and minimizing adverse effects. In pharmacology, for example, precise modeling of drug dynamics is vital to maximize therapeutic efficacy while minimizing patient harm, as in chemotherapy. However, current models, often developed by human experts, are limited by high cost, lack of scalability, and restriction to existing human knowledge. In this paper, we present the *Data-Driven Discovery (D3)* framework, a novel approach leveraging Large Language Models (LLMs) to iteratively discover and refine interpretable models of dynamical systems, demonstrated here with pharmacological applications. Unlike traditional methods, D3 enables the LLM to propose, acquire, and integrate new features, validate, and compare dynamical systems models, uncovering new insights into pharmacokinetics. Experiments on a pharmacokinetic Warfarin dataset reveal that D3 identifies a new plausible model that is well-fitting, highlighting its potential for precision dosing in clinical applications.

## 1 Introduction

The discovery of dynamical systems models plays a fundamental role across various domains, including pharmacology, epidemiology, and physical systems. In pharmacology, pharmacokinetic (PK) models are essential for understanding and predicting the time course of drug absorption, distribution, metabolism, and excretion in the body, which is crucial for optimizing therapeutic efficacy, minimizing toxicity, and personalized treatment regimens in diseases such as cancer, cardiovascular disorders, and infectious diseases [Gabrielsson and Weiner, 2001, Hedaya, 2012]. For example, cardiovascular disorders requiring Warfarin therapy affect tens of millions of individuals globally [Lee and Klein, 2013], highlighting the critical need for superior models to understand the dynamic impact of drugs and precision dosing, thereby assisting both doctors and patients.

The goal of PK modeling is to discover an underlying closed-form ordinary differential equation (ODE) $f$ from a dataset of observed patient trajectories. This problem is of significant interest to the machine learning (ML) community, as evidenced by previous non-interpretable ML modeling approaches aimed at developing better PK models [Chan and van der Schaar, 2022]. Such datasets are collected from expensive human clinical trials, necessitating the careful selection of which features

---

[*]Equal Contribution.
[†]Work done as a Postdoc at the University of Cambridge.

38th Conference on Neural Information Processing Systems (NeurIPS 2024).

to acquire and analyze during these trials [Guidance, 2010]. Pharmacometricians then leverage this data, using their existing knowledge to select appropriate pharmacokinetic models and employ standard statistical tools to infer the parameters of these models. For instance, they often fit a single compartmental PK model, a closed-form ODE model with 3-5 parameters, to the data [Chen and Abuassba, 2021]. This process is iterative, refining the models until the most accurate one that generalizes well to new patient trajectories is identified. However, this traditional model discovery approach is often ad-hoc and fundamentally limited by the human expert's time, experience, and implicit assumptions about the underlying pharmacokinetic processes.

Discovering interpretable pharmacokinetic models is traditionally performed by human experts through a scientific process of proposing, collecting, and validating models. Recently, there has been a growing call for artificial intelligence-driven methods to enhance pharmacological modeling [Ryan et al., 2024, Singh et al., 2023, Cheng et al., 2022]. This established process involves iterative steps, each presenting distinct challenges in constructing accurate pharmacokinetic dynamics models.

Selecting appropriate temporal models involves complex decisions on model class, state and feature variables, parameters, and their relationships, far exceeding the simplicity of standard pharmacometrics models like single, double, or triple compartmental models [Chen and Abuassba, 2021]. This complexity demands aligning model intricacy with data availability, where simple models suit small datasets and complex models fit larger ones. Acquiring the right features is crucial, as it must be done with limited prior information, unlike feature selection, which evaluates pre-existing features [Li et al., 2017]. The evaluation must identify inadequate models and explain why, requiring diverse and time-intensive tools beyond a single metric like validation MSE. These challenges are interrelated: strategic data acquisition and thorough evaluation are essential for effective modeling. Despite advancements in tackling these challenges independently, temporal modeling remains manual, limiting speed and scalability, underscoring the need for integrated and automated systems to enhance efficiency.

To address these challenges, we develop the *Data-Driven Discovery (D3)* framework, leveraging Large Language Models (LLMs) to iteratively discover and refine interpretable dynamics models, directly relevant to pharmacology, epidemiology and ecology applications. D3 features three agents: Modeling, Feature Acquisition, and Evaluation, which collaborate iteratively. The Modeling Agent uses LLMs' code generation and natural language understanding to explore the model space by generating hypothetical models. The Data Acquisition Agent utilizes LLMs' zero-shot and few-shot learning to optimize data acquisition based on summary statistics, text descriptions, and prior knowledge. The Evaluation Agent conducts comprehensive evaluations by computing refined validation MSE metrics, and alignment with prior process understanding, providing precise feedback to improve the other agents' performance.

**Contributions:** ① We propose the *Data-Driven Discovery (D3)* framework, a novel approach leveraging Large Language Models (LLMs) to iteratively discover and refine interpretable dynamics models, advancing pharmacokinetic modeling (Section 3). ② D3 overcomes the challenges of uncovering interpretable dynamical systems by using LLMs to explore vast model spaces and integrate unstructured data, producing models with few parameters that rival black box neural networks. D3 accurately discovers dynamics models across pharmacology, epidemiology, and ecology, matching the accuracy of existing methods while enhancing interpretability. It discovers a new, more accurate PK model for Warfarin, validated by expert pharmacometricians. We also gain insight into D3's ability to iteratively improve its models, selectively acquire features, and discover precise dynamics models.

## 2 The vast model space for temporal modeling

In this section, we introduce the model space that D3 searches through, which contains a diverse range of temporal models with varying degrees of refinement. Our focus is on Ordinary Differential Equation (ODE) models, which is one of the most widely used methods for modeling temporal dynamics, which include pharmacokinetic, physiological and epidemiological dynamics [Auger et al., 2008]. We are interested in modeling the evolution of a set of *state* variables $\mathbf{x}_n(t) \in \mathbb{R}^D$ for individuals $n = 1, \ldots, N$ over the time horizon $t \in [0, T]$. The state variables represent the target variables that the human experts are interested in modeling; and there may exist other feature variables that *can* help predict the evolution of state variables. We assume we are provided with a dataset of individuals trajectories such that $\mathcal{D} = \{(\mathbf{x}_n(t), \mathbf{a}_n(t)) \mid n = 1, \ldots, N, \ t \in [0, T]\}$, where $\mathbf{a}_n(t) \in \mathbb{R}^K$ denotes the observed individual-level features for individual. We can categorize ODE models into three levels (R1 - R3) based on their capacity for refinement.

**R1: Refinement through initial conditions $\mathbf{x}_n(0)$.** This level involves the simplest form of ODEs, commonly referenced in scientific literature [Schiesser, 2014]. These equations are defined over state variables $\mathbf{x}(t) \in \mathbb{R}^D$, incorporate global parameters $\theta$, and involve a function $f : \mathbb{R}^{D+1} \to \mathbb{R}^D$. The time $t$ spans the interval $[0, T]$:

$$\dot{\mathbf{x}}_n(t) = f(\mathbf{x}_n(t), t, \theta), \forall n \in [N] \tag{1}$$

Considering $N$ individuals, each represented by the trajectory $\mathbf{x}_n(t)$ for $n = 1, \ldots, N$, all follow the equation above. According to the uniqueness theorem for initial value problems in ODEs [Lindelöf, 1894], any differences among individuals' trajectories can be attributed solely to variations in the initial conditions $\mathbf{x}_n(0)$, under common regularity conditions on $f$.

**R2: Refinement through observed features $\mathbf{a}_n$.** Suppose we have access to a set of individual-level feature variables $\mathbf{a}_n$. We can enhance the model's refinement by incorporating these features into the ODE to enable unique dynamics for different individuals:

$$\dot{\mathbf{x}}_n(t) = f(\mathbf{x}_n(t), t, \mathbf{a}_n(t), \theta) \tag{2}$$

**R3: Refinement through acquired features $\mathbf{h}_n$.** Often, not all relevant variables are initially measured and available for analysis. In such cases, one may consider *acquiring* additional features $\mathbf{h}_n(t) \in \mathbb{R}^J$ to be integrated into the ODE. Here $\mathbf{h}_n(t)$ is a collection of $J$ features from the set of all *acquirable* features $\mathcal{H}$, where $J \leq |\mathcal{H}|$. This third level of refinement reflects one data acquisition challenge, i.e., identifying and collecting informative features $\mathbf{h}_n(t)$:

$$\dot{\mathbf{x}}_n(t) = f(\mathbf{x}_n(t), t, \mathbf{a}_n(t), \mathbf{h}_n(t), \theta) \tag{3}$$

After selecting the level of refinement (R1-R3), it is necessary to determine the functional form of the ODE $f$, which defines the interactions among various variables and parameters that govern the dynamics. There are two dominant approaches to parameterize $f$: using a concise closed-form white-box equation $f \in \mathcal{M}_C$ or employing a neural network $f \in \mathcal{M}_N$. The former approach is prevalent in the field of symbolic regression [Billard and Diday, 2002], while the latter is utilized in Neural ODEs [Chen et al., 2018]. Additionally, $f$ can incorporate both a closed-form component and a neural component, resulting in a hybrid ODE model. Nevertheless, all three approaches involve searching within large combinatorial spaces—closed-form equations, neural architectures, and both—presenting significant computational challenges.

# 3 Data-Driven Discovery (D3)

We aim to uncover the true underlying pharmacokinetic dynamical system, denoted as $f^*$. Our goal is to continually approximate $f^*$ as accurately as possible using proposed models $f_\theta$, with parameters $\theta$. For simplicity, we will refer to these models as $f$ in all subsequent references, omitting the $\theta$ term. To appropriately identify the refinement model level needed from the vast model space discussed above, we now introduce the *Data-Driven Discovery (D3)* framework, as depicted in Figure 1. This consists of three LLM agents: the Modeling Agent $G$, the Feature Acquisition Agent $A$, and the Evaluation Agent $E$. These three agents work together to form the model improvement loop and the data acquisition loop. The implementation details of the Agents are provided in Appendix F.

## 3.1 Inputs to Data-Driven Discovery (D3)

To begin the process, D3 requires a clearly defined description of the system in natural language, that details the modeling task, of creating a well-fitting model of *either* a white-box model or a hybrid model; that is a white-box model with a black-box neural network fitted to the residuals of the white-box model. D3 supports both discovering only white-box models and hybrid models, and we leave this choice up to the end user to decide for their application. Specifically, a user must provide a dataset of individual's trajectories $\mathcal{D}$, and a *system description* and the names and descriptions of any features that exist within the dataset, which can include the ranges of those features. Prior information can be added here by the practitioner into the system description if it is available. We now discuss how the three agents of Modeling, Feature Acquisition and Evaluation interact to discover iteratively better-fitting $f$ models. We provide full implementation details for the framework in Appendix F.

## 3.2 Modeling Agent $G$

D3's Modeling Agent $G$ is tasked to iteratively propose and refine the temporal model $f_i$, where $i = 1, \ldots, I$ indicates the iteration. We design $G$ following the three principles below.

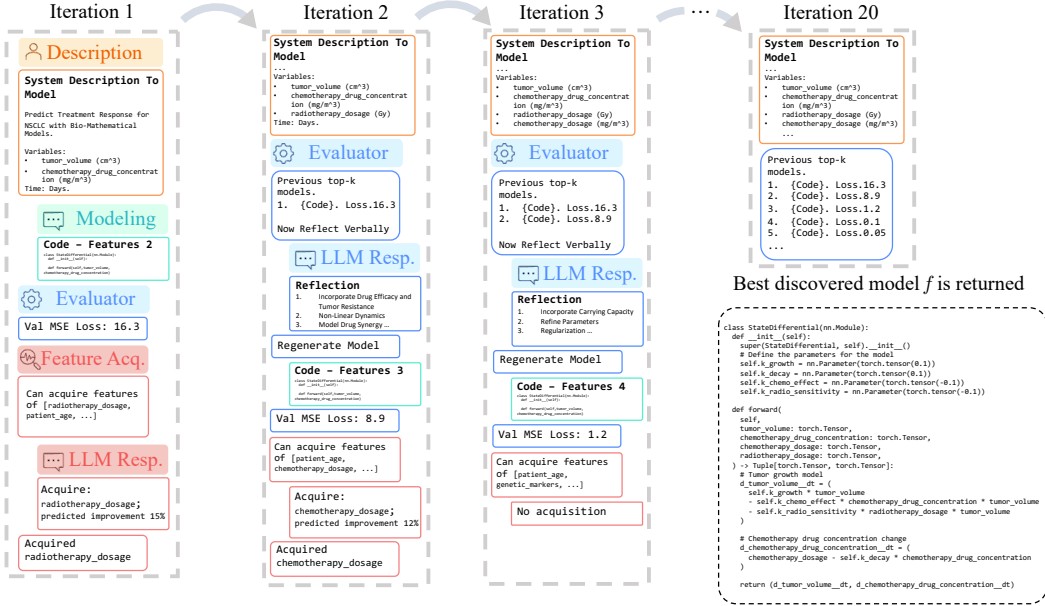

Figure 1: **Data-Driven Discovery (D3) Overview.** Given a dataset of trajectories of individuals $\mathcal{D}$, D3 can discover a well-fitting model $f$, that is either a white-box model or a hybrid model, combining a white-box model with a neural network component fit to the residuals. First, a user provides a system description $c$ for the model, which includes the feature names, their units and ranges. The Modeling agent uses the textual description and generates a model, represented as code. Next, the Evaluator agent evaluates the generated model on a held-out validation dataset and provides a loss metric. Next, the Feature Acquisition agent decides if it wants to acquire a new feature, that could aid in generating better models. This iterative process repeats, when there exists one or more previously generated models, the Evaluator agent provides a verbal reflection on how the model can be improved, which is used by the Modeling agent at subsequent iterations. This process repeats, discovering better fitting models, and after a set number of iterations, the best fitting model is returned, as code with its parameters optimized to the given training subset of the given dataset $\mathcal{D}$.

**Representing the model $f_i$ as code**. As we discussed in Section 1, one key challenge in evidence-based refinement is the vast space of possible ODE models, ranging from closed-form ODEs to complex neural Neural ODEs and hybrid models. As a prerequisite to finding the most suitable model, we need a consistent and flexible way to represent these diverse models such that they can be trained and validated on data. As such, D3 represents the model $f_i$ as code in Python language. Specifically, each model is defined as a class derived from 'nn.Module' in PyTorch [Paszke et al., 2017] that specifies the free parameters and the computation logic. The Modeling Agent $G$ leverages an LLM to generate the model code $f_i$ in each step.

**Informing generation with context $c$.** Effectively searching through the vast model space presents another challenge. Thankfully, in many applications there exists rich context $c$ about prior works and domain knowledge. However, this rich source of information was previously locked in unstructured documents and texts, making it hard for automated algorithms to make use of it. By leveraging the LLM's capability to understand unstructured documents, the Modeling Agent $G$ can now access and leverage this information, therefore proposing more informed models and searching more effectively. In this work, we consider the provision of the following categories of problem context of {system description} and {feature description} (Appendix F).

**Iterative model improvements based on rich verbal feedback $r_{i-1}$ and memory $s_{i-1}$.** To enhance the quality of the generated model $f_i$, we have designed the agent $G$ to iteratively improve the model based on the presence of any evaluation verbal feedback and its knowledge of the previously generated models and how they performed on a validation subset of given dataset $\mathcal{D}$ of trajectories. At each iteration $i$, the Evaluation Agent $E$ evaluates the previous model $f_{i-1}$ using the validation data subset of $\mathcal{D}_i$ and provides rich verbal feedback $r_{i-1}$ (c.f. Figure 1), which forms the foundation for the next generated model $f_i$. The Modelling Agent $G$ utilizes memory $s_{i-1}$ to track the top-k best-performing models so far, along with their associated evaluation feedback, here their respective losses. Further details on evaluation and data acquisition are discussed in the following section. This iterative process allows for continuous refinement and optimization of models based on their past performance and targeted feedback.

In summary, the Modeling Agent $G$ encapsulates a comprehensive range of model structures through computer code $f_i$ and efficiently explores the vast model space by integrating unstructured prior knowledge $c$, and utilizing data-informed feedback $r_{i-1}$ along with memory $s_{i-1}$. The operation of $G$ can be mathematically represented as follows:

$$f_i = G(c, f_{i-1}, \mathcal{D}_i, r_{i-1}, s_{i-1}) \tag{4}$$

### 3.3 Feature Acquisition Agent $A$

D3's Feature Acquisition Agent $A$ is tasked with iteratively proposing new features for acquisition. These features are selected based on their potential to improve the next model's performance in modeling the target state $\mathbf{x}_n(t)$.

$$h_i = A(c, f_i, \mathcal{D}_i, r_{i-1}, s_{i-1}) \tag{5}$$

Where $h_i$ is the next feature to acquire. Specifically, when a feature is acquired we make the assumption, that it is acquired for all individuals, therefore at the next iteration the dimension of the individual-level features $\mathbf{a}_n(t) \in \mathbb{R}^K, n = 1, \ldots, N$ increases by one, to $\mathbf{a}_n(t) \in \mathbb{R}^{K+1}, n = 1, \ldots, N$. It is possible to do this in clinical trials, where pharmacokinetic datasets of patients are collected [Dziura et al., 2013], for example acquiring the individual's ages, or other lab biomarker measurements or tests [Derraik et al., 2021].

**Estimating the value of the feature**. We leverage the value of information [Feltham, 1968] framework to provide a principled way to determine which feature, if any to acquire next. This follows as:

$$V(h_i) = \mathbb{E}[L(f_i, h_i, \mathcal{D}_{i-1})] - \mathbb{E}[L(f_i, \mathcal{D}_{i-1})] - l(h_i), \tag{6}$$

where the first two terms capture the improvements in validation loss $L$ when model $f_i$ is trained on the new dataset that includes the new feature, that is $\mathcal{D} = \{(\mathbf{x}_n(t), \mathbf{a}_n(t), \mathbf{h}_n(t)) | \forall n \in [N]\}$ and $l(h_i)$ represents the cost for acquiring $h_i$. Estimating $V(h_i)$ presents a statistical challenge and a computational challenge. Firstly, in practice, we often do not have access to $h_i$ to begin with (as the goal is to acquire *new* features)—hence we need a way to estimate $\mathbb{E}[L(f_i, h_i, \mathcal{D}_{i-1})]$ without fully accessing $h_i$. Secondly, even when $h_i$ is available, computing the two loss terms involves fitting models twice which can be computationally challenging.

To address these challenges, the Feature Acquisition Agent $A$ *predicts* the value of feature $V(h_i)$ from the available information about $h_i$, as follows:

$$\hat{V}(h_i) = g(c_{h_i}, T(h_i), \mathcal{D}_i), \tag{7}$$

where $c_{h_i}$ represents the unstructured prior information about the feature $h_i$ (e.g. data descriptions and metadata), $T(h_i)$ represents the available summary statistics (e.g. range and type of values from a small subset of data $h_i$), $\mathcal{D}_{i-1}$ represents the existing data, and $g$ is the prediction function.

The prediction challenge above corresponds to zero-shot or few-shot learning in ML. In the zero-shot case, we need to predict $\hat{V}(h_i)$ without having access to any "label" $V(h)$, for some feature $h$. In the few-shot case, we have access to a small number of feature-label pairs, $\{c_{h_i}, T(h_i), h_{i-1}, V(h_{i-1})\}$ to learn the prediction function $g$. The Feature Acquisition Agent $A$ leverages LLM's capability in zero-shot and few-shot learning to address this estimation challenge. Specifically, we inform the LLM by providing high-level statistics $T(h_i)$ and a description of the feature $c_{h_i}$ in the form of the feature name to warm-start the zero-shot learning. When the $h_i$ has been included in the data, we inform the LLM about the validation loss associated with those data $v(h_i)$ and add it to the few-shot examples to guide the estimation.

**Context, feedback and memory**. Similar to $G$, $A$ is also able to leverage unstructured problem context $c$, evaluation feedback $r_{i-1}$, and memory $s_{i-1}$ to guide the data acquisition proposal.

### 3.4 The Evaluation Agent $E$

The Evaluation Agent $E$ is vital for iteratively improving the Modeling Agent $G$ and Feature Acquisition Agent $A$. It provides feedback $r_i$:

$$r_i = E(c, f_i, \mathcal{D}_i) \tag{8}$$

Feedback $r_i$ can be numeric, such as validation loss or mean squared error (MSE) of the model $f_i$ on the validation subset $\mathcal{D}_i$, or more refined metrics like validation loss per target dimension in $D$. It can

also be textual, offering detailed suggestions for model improvements, including clinically plausible modifications generated by an LLM [Shinn et al., 2024].

Evaluating dynamical systems is complex due to intricate interactions and temporal dependencies. The Evaluation Agent dynamically assesses both model *performance* and *plausibility*, ensuring models are both accurate and interpretable. This comprehensive evaluation is crucial for developing sophisticated, clinically relevant pharmacokinetic models that address real-world complexities.

## 4 Related Works

Table 1: Comparison with related works in addressing the Modeling and Data Acquisition Challenges. An empty field means not applicable. *Refinement Level*: the level of refinement (R1 - R3) the method can capture. *Class of* $f$: the class of ODE model $f$ (closed-form $\mathcal{M}_C$, neural networks $\mathcal{M}_N$). *Context* $c$: whether the method leverages unstructured contextual information to guide search. *Sample* $N$: whether the method is able to acquire new samples. *Feature* $\mathbf{h}$: whether the method can acquire new feature variables. *Goal*: whether the goal of data acquisition is to improve training, evaluation, or both.

| | Modeling Challenge | | | Data Acquisition Challenge | | |
|---|---|---|---|---|---|---|
| Method | Refinement Level | Class of $f$ | Context $c$ | Zeroshot | Feature $\mathbf{h}$ | Goal |
| Symbolic Reg. | R1, R2 | $\mathcal{M}_C$ | $\times$ | | | |
| Neural ODE | R1, R2 | $\mathcal{M}_N$ | $\times$ | | | |
| AI Feynman | R1 | $\mathcal{M}_C$ | $\checkmark$ | | | |
| Eureka | R1, R2 | $\mathcal{M}_C$ | $\checkmark$ | | | |
| AFA | | | | $\times$ | $\checkmark$ | Train |
| Active Learning | | | | $\times$ | $\times$ | Train |
| Active Testing | | | | $\times$ | $\times$ | Eval |
| D3 | R1 - R3 | $\mathcal{M}_C \cup \mathcal{M}_N$ | $\checkmark$ | $\checkmark$ | $\checkmark$ | Both |

Our work focuses on autonomously learning temporal models while acquiring data, with several relevant research strands, as summarized in Table 1, which is expanded in Appendix A.

**ODE learning methods.** Symbolic regression methods like SINDy and D-CODE [Koza, 1994, Brunton et al., 2016, Qian et al., 2022] can discover closed-form ODEs $f \in \mathcal{M}_C$ using genetic algorithms. They can incorporate features $\mathbf{a}_n$ (R2 refinement) but struggle with many variables (e.g., >20) due to computational complexity. Neural ODEs use neural networks $f \in \mathcal{M}_N$ to handle many variables [Chen et al., 2018, Dupont et al., 2019, Zaytar and El Amrani, 2016, Devlin et al., 2018, Sehovac and Grolinger, 2020], but have numerous free parameters, risking overfitting with insufficient data. D3 captures $f \in \mathcal{M}_C \cup \mathcal{M}_N$ and supports R1 - R3 refinement, adapting based on data availability. Both Symbolic regression and Neural ODEs require manual context incorporation $c$ via hyperparameters and do not address Data Acquisition Challenges.

**AI for automated modeling.** AI Feynman [Udrescu and Tegmark, 2020] enhances symbolic regression by using the physical units of variables, which provide additional constraints to narrow the search space. However, its utility outside physics is limited, as units in other fields carry less information. Eureqa [Ma et al., 2023] applies LLMs to model the reward function in reinforcement learning, integrating unstructured context and representing models as code. Unlike D3, Eureqa cannot automatically refine models, acquire new data, or easily apply to learning temporal dynamics like ODEs.

**Data Acquisition.** Methods like Active Feature Acquisition (AFA) [Ma et al., 2018, Gong et al., 2019] measure additional features on existing samples to improve performance. Active Learning [Sebastiani and Wynn, 2000, Settles, 2009, Sener and Savarese, 2017, Imberg et al., 2020] and Active Testing [Lowell et al., 2018, Kossen et al., 2021] acquire new samples to increase training or evaluation data size while keeping features constant. These methods do not address the zero-shot setting, where data acquisition decisions must be made before observing any target data (Equation 7). Additionally, most Active Learning methods focus on acquiring labels for supervised learning, which is not directly applicable to temporal modeling tasks.

## 5 Experiments and Evaluation

In this section, we demonstrate that D3 can discover well-fitting pharmacokinetic dynamical system models for a range of diverse PK datasets, including an epidemiological and ecological dataset.

**Benchmark Datasets**. Our evaluation encompasses six real-world datasets with clinical relevance, each originating from either real-world data or highly accurate simulators developed by human experts.

Three datasets are based on a state-of-the-art biomedical Pharmacokinetic-Pharmacodynamic (PKPD) model of lung cancer tumor growth, which simulates the combined effects of chemotherapy and radiotherapy in lung cancer [Geng et al., 2017] (Equation (11)). This model has been widely utilized in previous research [Bica et al., 2020, Seedat et al., 2022, Melnychuk et al., 2022]. Specifically, we employ this bio-mathematical model to generate three variations: lung cancer without treatment (**Lung Cancer**), lung cancer treated with chemotherapy (**Lung Cancer (with Chemo.)**), and lung cancer treated with both chemotherapy and radiotherapy (**Lung Cancer (with Chemo. & Radio.)**). Additionally, we utilize an intricate COVID-19 epidemic agent-based simulator (**COVID-19**) [Kerr et al., 2021], to provide an epidemiological dataset. Another dataset comes from an ecological model simulating a microcosm of algae, flagellate, and rotifer populations (**Plankton Microcosm**), replicating a three-species prey-predator experimental system [Hiltunen et al., 2013]. Finally, we include a real Pharmacokinetic (PK) dataset of Warfarin patients (**Warfarin**) [Janssen et al., 2022]. Detailed information about all benchmark datasets is provided in Appendix B.

**Benchmark Methods**. To evaluate the performance of D3 and establish its competitive performance, we conduct comparisons with leading modeling methods for ODEs. Specifically, we benchmark against advanced black-box models that have many parameters, such as neural ODEs with action inputs, known as **DyNODE** [Chen et al., 2018, Alvarez et al., 2020]. Also in this class are recurrent neural networks (**RNN**) and a state-of-the-art transformer model (**Transformer**). In addition, we include white-box transparent dynamical systems models identified through equation discovery techniques, such as Sparse Identification of Nonlinear Dynamics (**SINDy**) [Brunton et al., 2016]. Moreover, D3 supports two discovery modes, of which we compare against both, discovering only white-box models (**D3-white-box**) and discovering hybrid models (**D3-hybrid**)[3]. We also perform ablations of D3, of zero-shot generated model from D3 as (**ZeroShot**) and the same model with optimized parameters (**ZeroOptim**). Detailed descriptions of the implementations, hyperparameters, and experimental procedures for these benchmarks can be found in Appendix E.

**Evaluation Metrics**. To assess the performance of our benchmark methods, we use the mean squared error (MSE) on a held-out test dataset of state-action trajectories. This evaluation is conducted over ten runs, each initialized with different random seeds. We report the average MSE from these runs along with their 95% confidence intervals. Further details can be found in Appendix C.

# 6 Main Results

We conducted a comprehensive evaluation of our benchmark methods across all datasets, as tabulated in Table 2. We observe that D3 can discover well-fitting dynamical system models, achieving low mean squared error in test predictions on the held-out test dataset of individual trajectories. Crucially, it can discover concise closed-form equation white-box models, of 5-15 parameters that can outperform some of the standard white-box and black-box modeling methods, indicating it is discovering well-fitting underlying equations for the respective systems of interest.

Table 2: **Evaluating Method Performance.** We report the test prediction mean squared error (MSE) on held-out datasets across benchmarks. D3 consistently has the lowest error. Results are averaged over ten random seeds with 95% confidence intervals.

| Method | Lung Cancer MSE ↓ | Lung Cancer (with Chemo.) MSE ↓ | Lung Cancer (with Chemo. & Radio.) MSE ↓ | Plankton Microcosm MSE ↓ | COVID-19 MSE ↓ | Warfarin PK MSE ↓ |
|---|---|---|---|---|---|---|
| DyNODE | 326±5.96 | 55.7±52.8 | 16.2±6.35 | 0.000397±0.000883 | 74±2.69 | 0.726±0.17 |
| SINDy | 325±5.95 | 11.8±0.442 | 13.7±0.635 | 0.00135±0 | 93.5±0.509 | 6.84±1.76 |
| ZeroShot | 5.78e+03±7.6e+03 | 304±86.1 | 6.44e+03±4.27e+03 | 0.333±0.274 | 2.47e+03±2.52e+03 | 1.81±8.53 |
| ZeroOptim | 225±204 | 33.8±50.8 | 6.38±8.97 | 0.0133±0.0013 | 7.88±0.0468 | 398±5.05e+03 |
| RNN | 1.16e+06±3.21e+04 | 719±94.3 | 137±5.88 | 0.0306±0.0459 | 1.39e+04±2.47e+03 | **0.0495±0.0406** |
| Transformer | 7.07±0.558 | 0.346±0.0701 | 0.207±0.0318 | 3.42e-05±1.97e-05 | **0.261±0.0915** | 1.33±0.941 |
| **D3-white-box** | 59.4±101 | 4.8±11.8 | 2.42±2.02 | 0.000245±0.00022 | 5.92±1.17 | 19.6±40.3 |
| **D3-hybrid** | **4.72±9.16** | **0.0978±0.0463** | **0.135±0.225** | **1.86e-06±1.87e-06** | 1.88±2.57 | 0.647±0.167 |

## 6.1 Case Study: Discovering Novel PK Models for Warfarin

PK models of warfarin, a widely used anticoagulant, are crucial due to its extensive use in treating deep vein thrombosis, pulmonary embolism, and stroke prevention in atrial fibrillation and patients with mechanical heart valves, with millions of prescriptions globally [Lee and Klein, 2013]. We applied D3 to a public dataset of 33 patient trajectories [Janssen et al., 2022], discovering a novel PK model for warfarin that outperforms existing literature (Table 3). We compare the test MSE of the

---

[3]Code is available at `https://github.com/samholt/DataDrivenDiscovery` and we provide a broader research group code base at `https://github.com/vanderschaarlab/DataDrivenDiscovery`.

discovered model against existing models and provide pharmacologist feedback. Full model details and results from running D3-hybrid on the same dataset are in Appendix G.

**Existing Wafarin PK Model.** The standard pharmacokinetic (PK) warfarin model from the literature [Lv et al., 2017, Hamberg, 2013] is the following:

$$\frac{dC}{dt} = k_a \cdot D - k_e \cdot C,$$
$$k_e = k_{e,\text{base}} + k_{e,\text{age}} \cdot A + k_{e,\text{sex}} \cdot S \qquad (9)$$

Table 3: Warfarin Modeling Comparison

| Method | Warfarin Best Model Test MSE |
|---|---|
| Existing Warfarin PK | 0.646 |
| **D3-white-box** | **0.39** |
| **D3-hybrid** | **0.271** |

where $C$ is the concentration of warfarin, $D$ is the dosage administered, $A$ represents the age of the patient, $S$ denotes the sex of the patient (1 for male, 0 for female), $k_a$ is the absorption rate, $k_{e,\text{base}}$ is the base elimination rate, $k_{e,\text{age}}$ represents the decrease in elimination rate per year increase in age, and $k_{e,\text{sex}}$ is the difference in elimination rate between sexes. This standard model achieves a test loss of 0.646. Whereas D3 can discover both a white-box model with a test loss of 0.39 and a hybrid model with a test loss of 0.271, which both outperform the standard model. Of particular relevance is the white-box model as it is fully interpretable by pharmacometricians, whilst still being a precise model.

**New Discovered PK Warfarin Model.** D3-white-box discovered a new warfarin PK white-box model with a test loss of 0.39, of the following:

$$\frac{dC}{dt} = \sqrt{D} - k_{\text{eff}} \cdot \frac{C}{K_m + C},$$
$$k_{\text{eff}} = k_{e,\text{base}} + k_{e,\text{age}} \cdot (A - \overline{A}) + k_{e,\text{sex}} \cdot (S - \overline{S})$$
$$+ k_{\text{decay}} \cdot C + k_{ds} \cdot D \cdot (S - \overline{S}) \qquad (10)$$
$$+ k_{as} \cdot (A - \overline{A}) \cdot (S - \overline{S}) + k_{ad} \cdot D \cdot (A - \overline{A})$$

where the additional parameters $k_{\text{decay}}$ represent the natural decay rate of warfarin concentration and $K_m$ is the Michaelis constant indicating the warfarin concentration at which the metabolism rate is half its maximum. The parameters $k_{ds}$, $k_{as}$, and $k_{da}$ are the interaction terms for dosage-sex, age-sex, and dosage-age, respectively, and $\overline{A}$ and $\overline{S}$ are the sample population means for the covariates of age and sex. This new PK model includes the original parameters along with additional interaction terms and transformations, enhancing the model's complexity and accuracy.

The model discovered by D3 introduces key innovations in warfarin pharmacokinetics. *Square Root Transformation for Dosage Effect*: This non-linear transformation moderates high doses, enhancing sensitivity to dosage variations, unlike standard models. *Natural Decay Term*: Proportional to the current warfarin concentration, it better reflects elimination kinetics and improves time-based predictions. *Michaelis-Menten Saturation Kinetics*: Addresses metabolic pathway saturation at higher concentrations, crucial for warfarin. *Interaction Terms (dosage-sex, age-sex, dosage-age)*: Capture complex interactions between patient-specific factors, providing a nuanced understanding of drug dynamics. These advancements make D3 a powerful tool for enhancing pharmacokinetic predictions in clinical settings. The discovered hybrid model is detailed in Appendix G.

**Expert Clinical Commentary.** We sought feedback from expert pharmacologists on the discovered model[4]: *Prof. Eoin McKinney, Clinician*. "This model is significant, as consortiums are dedicated to improving Warfarin modeling [Consortium, 2009]. The model adds novel components, such as the Michaelis component for time-varying changes and novel interaction terms like age-sex." *Jean-Baptiste Woillard, Pharmacologist*. "The model is promising and pharmacokinetically plausible. The next step is to apply D3 to other clinically relevant PK drug datasets." *Richard Peck, Clinical Pharmacologist*. "This model is reasonable and potentially superior. It represents a significant advance in clinical pharmacology by automatically identifying robust PK models."

## 6.2 Insight Experiments

This section provides an in-depth analysis of D3's effectiveness related to its benchmark counterparts.

**Can D3 perform feature acquisition and leverage the LLM prior information to perform this better?** To explore the feature acquisition performance of D3, we showcase this component working

---

[4]We provide full commentary in Appendix D.

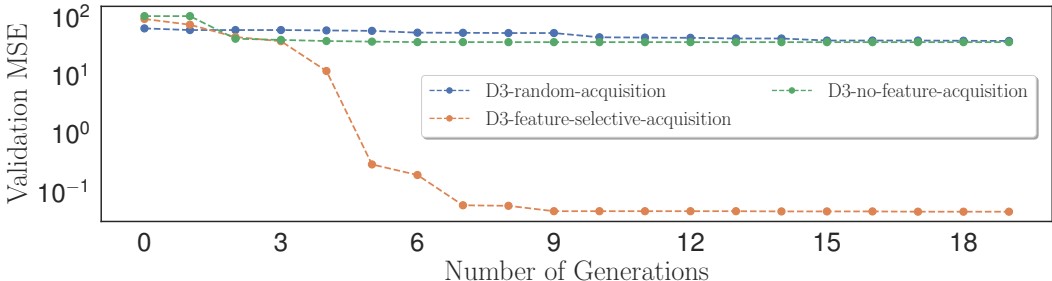

Figure 2: **Feature Acquisition.** D3 performing adaptive feature acquisition in the Lung Cancer (with Chemo. & Radio.) dataset. We observe that D3 still achieves the lowest test prediction error.

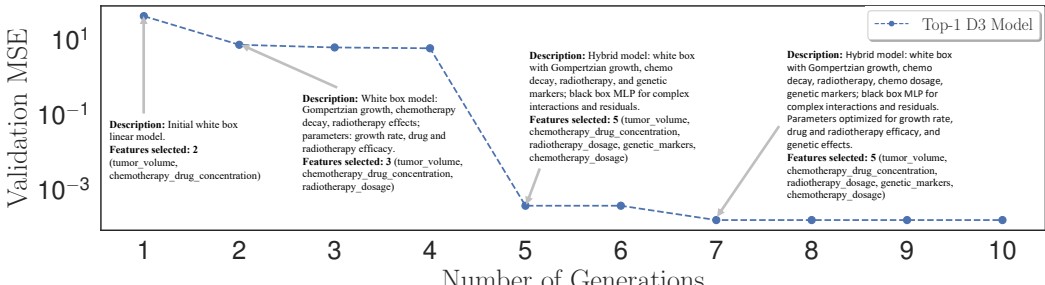

Figure 3: **D3 iteratively discovers better models** $f_i$. Validation MSE of the model generated in each iteration, showing the best-generated model (Top-1 model)—additionally with a few of the models labeled with their model descriptions, and features acquired at the generation. D3 can effectively acquire and integrate new features, validate, and compare models to achieve a better-fitting model.

in full, by comparing it against the baseline of a random feature acquirer policy, a null policy that just uses the existing features and our approach of D3 that leverages the LLM to quantify the value of information for features, where no training data exists for them in the dataset. We observe, as in Figure 2, that D3 converges the fastest, and achieves the overall highest performance, whereas the other feature acquisition methods fall short. Experimental details are in Appendix G.2.

**Can D3 evolve its modular model to fit the system best?** We analyze this from an empirical point of view to determine if D3 can correctly evolve the generated model and reduce its prediction error over subsequent generations. We observe that D3 can indeed understand, reason, and iterate the generated code representation of the model to incorporate a better fitting model, as observed in Figure 3.

# 7 Conclusion

**Summary**. In this paper, we proposed *Data-Driven Discovery (D3)* framework, to iteratively discover and refine interpretable models of pharmacological dynamics, where it has the ability to acquire and integrate new features, validate, and compare pharmacological dynamical systems models, thereby uncovering new insights into pharmacokinetic and physiological processes. Specifically, applied to a real Warfarin PK dataset we were able to discover a better performing Warfarin PK model and provide new insights into what an optimal PK model for Warfarin may need to possess, such as additional Michaelis components.

**Limitations & Future work**. There exist limitations to the current approach. First, the LLM discovery framework is an initial framework, and the utility of the LLM to generate better models could be enhanced, tree-based generation strategies such as the tree of thought [Yao et al., 2024], or graph of thought [Besta et al., 2024]. Specifically, it relies on a capable enough LLM, that can use tools, where we specifically used GPT4 as the underlying LLM. Moreover, we make the assumption that when D3 acquires a new feature it acquires that feature for all the individuals within the existing dataset, we leave for future work to consider applications where this assumption may not always be true. Furthermore, we provide to the LLM a system description from a user, future works could explore avoiding the user to provide this, and instead leveraging a form of automatic retrieval augmented generation [Lewis et al., 2020].

**Broader Impact and Ethical Considerations**. Principally D3 aims to discover interpretable models of pharmacological dynamics. However, the final discovered models should always be checked

by appropriate human experts and validated in additional held-out datasets before any clinical use. Furthermore, D3 is a tool, which could be misused by a malicious user with unethical system descriptions as input to discover a potentially biased model. Moreover, D3 leverages LLMs and their feedback as an integral component, however, LLMs are prone to hallucinations, thereby motivating any LLM-generated outputs that are shown to the user should have a content filter applied to them.

## Acknowledgments and Disclosure of Funding

We extend our gratitude to the anonymous reviewers, area and program chairs, members of the van der Schaar lab, and Andrew Rashbass for their valuable feedback and suggestions. SH gratefully acknowledges the sponsorship and support of AstraZeneca. ZQ did this work whilst a postdoc in the van der Schaar lab. This work was funded by Microsoft's Accelerate Foundation Models Academic Research initiative.

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

# Appendix

## Table of Contents

# A  Additional Related Work

We expand on the related work contained within the main paper.

**ODE learning methods.** Symbolic regression methods [Koza, 1994, Kacprzyk et al., 2024, Holt et al., 2023b] such as SINDy and D-CODE [Brunton et al., 2016, Qian et al., 2022] can automatically discover closed form ODEs $f \in \mathcal{M}_C$ through genetic algorithms. In principle, they can incorporate features $\mathbf{a}_n$, achieving refinement R2, but they struggle with many variables (e.g., more than 20) due to computational complexity. On the other hand, Neural ODE leverages neural networks $f \in \mathcal{M}_N$ to capture a large number of variables [Chen et al., 2018, Dupont et al., 2019, Zaytar and El Amrani, 2016, Devlin et al., 2018, Sehovac and Grolinger, 2020, Holt et al., 2022, 2023a, 2024a]. However, Neural ODEs involve many free parameters and cannot automatically reduce the level of refinement when there is insufficient training data, leading to overfitting. In comparison, D3 can capture $f \in \mathcal{M}_C \cup \mathcal{M}_N$ and R1 - R3 levels of refinement; it can further adapt the level of refinement based on data availability. Furthermore, in both Symbolic regression and Neural ODE, human experts need to manually incorporate the problem context $c$ through the specification of many hyperparameters. Finally, neither of these methods assumes fixed training data and does not address the Data Acquisition Challenges.

**AI for automated modeling.** AI Feynman [Udrescu and Tegmark, 2020] improves upon the symbolic regression methods by leveraging a specific type of information $c$, i.e. the physical unit of each variable (e.g., meter). In applications in physics, knowing the units provides additional constraints on the variable's relationship, thus helping narrow down the search space. However, its utility outside physical sciences has been limited. More recently, Eureqa [Ma et al., 2023] applies LLMs to modeling the reward function of reinforcement learning algorithms. Eureqa is similar to D3 in its ability to integrate unstructured context $c$ and represent the model as code. However, Eureqa cannot automatically adjust the level of refinement or acquire new data and it does not easily apply to learning temporal dynamics, e.g. ODEs. Furthermore, other LLM code generation [Holt et al., 2024b,c] approaches cannot acquire new data.

**Data Acquisition.** Methods have been developed to address specific data acquisition challenges. Active Feature Acquisition (AFA) attempts to measure additional feature variables $\mathbf{h}$ on existing training samples to improve predictive performance [Ma et al., 2018, Gong et al., 2019]. On the other hand, Active Learning [Sebastiani and Wynn, 2000, Settles, 2009, Sener and Savarese, 2017, Imberg et al., 2020] and Active Testing [Lowell et al., 2018, Kossen et al., 2021] attempt to acquire new samples to increase the size of training or evaluation data $N$ while keeping the features constant. Neither field has proposed a method to address the *zero-shot* setting, where the data acquisition decision must be made before observing any target data (Equation 7). Moreover, most existing methods in Active Learning focus on acquiring labels for supervised learning given unlabelled samples, which is not immediately applicable to temporal modeling tasks.

# B  Benchmark Dataset Environment Details

In the following, we present six clinically relevant datasets, each of which is either sourced from real-world data or generated from a high-fidelity simulator developed by domain experts.

## B.1  Cancer PKPD

Three of our environments are derived from a state-of-the-art biomedical Pharmacokinetic-Pharmacodynamic (PKPD) model of lung cancer tumor growth, used to simulate the combined effects of chemotherapy and radiotherapy in lung cancer [Geng et al., 2017]. This model has been extensively used in other works [Seedat et al., 2022, Bica et al., 2020, Melnychuk et al., 2022]. We use this bio-mathematical lung cancer model to create three variations: no treatments (**Lung Cancer**), chemotherapy only (**Lung Cancer (with Chemo.)**), and both chemotherapy and radiotherapy (**Lung Cancer (with Chemo. & Radio.)**). For each model, we sample a respective dataset. Below, we detail the general case of *Lung Cancer (with Chemo. & Radio.)*, which originates from the general Cancer PKPD Model, and then describe the variations.

**Cancer PKPD Model**. This model simulates the combined effects of chemotherapy and radiotherapy on lung cancer tumor growth [Geng et al., 2017], as shown in Equation (11). The model predicts

tumor volume $x(t)$ over time $t$ in days post-diagnosis. The model includes two binary treatments: (1) radiotherapy $u_t^r$ and (2) chemotherapy $u_t^c$.

$$\frac{dx(t)}{dt} = \Big( \underbrace{\rho \log \Big( \frac{K}{x(t)} \Big)}_{\text{Tumor growth}} - \underbrace{\beta_c C(t)}_{\text{Chemotherapy}} - \underbrace{(\alpha_r d(t) + \beta_r d(t)^2)}_{\text{Radiotherapy}} \Big) x(t) \tag{11}$$

The parameters $K, \rho, \beta_c, \alpha_r, \beta_r$ for each simulated patient are specified in Geng et al. [2017] and summarized in Table 4. Additionally, the chemotherapy drug concentration $C(t)$ follows an exponential

Table 4: **Cancer PKPD parameter values.**

| Model | Variable | Parameter | Parameter Value |
|---|---|---|---|
| Tumor growth | Growth parameter | $\rho$ | $7.00 \times 10^{-5}$ |
| | Carrying capacity | $K$ | 30 |
| Radiotherapy | Radio cell kill ($\alpha$) | $\alpha_r$ | 0.0398 |
| | Radio cell kill ($\beta$) | $\beta_r$ | Set such that $\alpha/\beta = 10$ |
| Chemotherapy | Chemo cell kill | $\beta_c$ | 0.028 |

decay with a half-life of one day:

$$\frac{dC(t)}{dt} = -0.5C(t) \tag{12}$$

The chemotherapy binary action represents increasing the $C(t)$ concentration by $5.0\text{mg/m}^3$ of Vinblastine given at time $t$. The radiotherapy concentration $d(t)$ represents 2.0 Gy fractions of radiotherapy given at timestep $t$, where Gy is the Gray ionizing radiation dose.

**Time-dependent confounding.** We introduce time-varying confounding by modeling chemotherapy and radiotherapy assignment as Bernoulli random variables. The probabilities $p_c$ and $p_r$ depend on tumor diameter as follows:

$$p_c(t) = \sigma \left( \frac{\gamma_c}{D_{\max}} (\bar{D}(t) - \delta_c) \right) \qquad p_r(t) = \sigma \left( \frac{\gamma_r}{D_{\max}} (\bar{D}(t) - \delta_r) \right), \tag{13}$$

where $D_{\max} = 13\text{cm}$ is the maximum tumor diameter, $\delta_c = \delta_r = D_{\max}/2$, and $\bar{D}(t)$ is the average tumor diameter. The parameters $\gamma_c$ and $\gamma_r$ control the extent of time-varying confounding, with $\gamma_c = \gamma_r = 2$.

**Sampling datasets.** Using the Cancer PKPD model, we sample $N = 1,000$ trajectories, corresponding to $N = 10,000$ patients. Initial tumor volumes are sampled from a uniform distribution $x(0) \sim \mathcal{U}(0, 1149)$, and patient trajectories are forward simulated for 60 days using the Cancer PKPD Equation (11) and the action policy of Equation (13), employing a Euler stepwise solver. This process generates one dataset sample. We repeat this with independent random seeds to create $\mathcal{D}_{\text{train}}, \mathcal{D}_{\text{val}}, \mathcal{D}_{\text{test}}$. For each benchmark method run with a random seed, we re-sample the datasets. Each variation includes either chemotherapy, both chemotherapy and radiotherapy, or neither. We provide further details of this dataset's system description and variable descriptions in Appendix F.5.

### B.2 COVID-19

We utilize the sophisticated epidemic agent-based simulator COVASIM [Kerr et al., 2021] to model COVID-19 epidemics. This advanced simulator is capable of simulating both non-pharmaceutical interventions (such as lockdowns, social distancing, and school closures) and pharmaceutical interventions (such as vaccinations). In this agent-based simulator, each agent represents an individual within the population, and can be in one of several states: susceptible to COVID-19, exposed, infectious, or recovered (including deaths).

We use COVASIM with its default parameter settings provided by the open-source implementation [5]. COVASIM simulates a population of individuals, and to ensure accuracy, we simulate 24 countries,

---

[5]COVASIM is an open-source simulator, available at `https://github.com/InstituteforDiseaseModeling/covasim`.

collecting trajectories for each. For each simulation, we use a population size of $1,000,000$ individuals, simulating each individual separately (disabling simulation rescaling). The simulation begins with a random number of individuals initially infected with COVID-19, $I(0) = \mathcal{U}(10,000, 100,000)$, and proceeds for 60 days.

We repeat this process with independent random seeds to generate $\mathcal{D}_{\text{train}}, \mathcal{D}_{\text{val}}, \mathcal{D}_{\text{test}}$. For each benchmark method run with a different random seed, we re-sample the datasets. Detailed descriptions of this dataset's system and variables are provided in the prompt template outlined in Appendix F.5.

### B.3 Plankton Microcosm

This subsection describes an ecological model of a microcosm consisting of algae, flagellate, and rotifer populations, replicating an experimental three-species prey-predator system [Hiltunen et al., 2013]. We use the dataset provided by [Bonnaffé and Coulson, 2023][6]. The dataset contains a single trajectory of 102 time steps. We split the data into training, validation, and test sets with proportions of 70%, 15%, and 15%, respectively, ensuring that the splits maintain the chronological order to preserve temporal causality.

Detailed descriptions of this dataset's system and variables are provided in the prompt template outlined in Appendix F.5.

### B.4 PK Wafarin Dataset

Here we describe the real PK Wafarin dataset from a clinical trial that is publicly available [Janssen et al., 2022]. We use the NOMEN dataset available at `https://github.com/Janssena/SI-AIEP-paper`. This publicly available dataset consists of 32 patients who received warfarin in a clinical trial, which was originally designed to determine how to predict drug concentrations in patients. The dataset includes a total of 251 warfarin concentration measurements, with a median of six measurements per patient. Each patient received a single dose of warfarin at $t = 0$, and measurements were taken at $t \in \{0.25, 0.5, 1.0, 2.0, 4.0, 6.0, 12.0, 24.0, 48.0, 72.0, 96.0, 120.0\}$. The available covariates in the dataset are patient weight, age, and sex. We follow the dataset's original pre-processing code, which is provided at `https://github.com/Janssena/SI-AIEP-paper`. We split the data into training, validation, and test sets with proportions of 70%, 15%, and 15%, respectively, ensuring that the splits maintain the chronological order to preserve temporal causality. This dataset is released under a GPL-3.0 license.

Detailed descriptions of this dataset's system and variables are provided in the prompt template outlined in Appendix F.5.

## C Evaluation Metrics

We utilize mean squared error (MSE) to assess the benchmark methods on a separate test dataset comprising individual trajectories, denoted as $\mathcal{D}_{\text{test}}$. This assessment is based on the loss defined in Equation (14) and reported as MSE. The metrics are averaged over ten runs with different random seeds, and we provide these averages along with their 95% confidence intervals. Specifically we provide the mean confidence interval[7]. For each random seed, we independently generate new training, validation, and test datasets when a simulator is available. Specifically, for each simulation, we ensure that the validation and test datasets contain the same number of trajectories as the training set. Each baseline model is trained on the training dataset, with early stopping applied using the validation dataset when supported by the method. Subsequently, we evaluate the performance of each baseline on the test dataset. This entire procedure is repeated for each random seed. All experiments and training were conducted using a single Intel Core i9-12900K CPU @ 3.20GHz, 64GB RAM, and an Nvidia RTX3090 GPU with 24GB of memory.

---

[6]The Plankton Microcosm and Hare-Lynx datasets are both open source and available at `https://github.com/WillemBonnaffe/NODEBNGM`.

[7]We use the following code `https://stackoverflow.com/questions/15033511/compute-a-confidence-interval-from-sample-data`.

## C.1 Model Optimization Losses

We evaluate the optimization loss using mean squared error (MSE) on a dataset $\mathcal{D}$, and also consider a component-wise MSE loss for more detailed analysis.

**MSE Loss.** Specifically, we optimize the following mean squared error objective:

$$\mathcal{L}(\theta, \mathcal{D}) = \frac{1}{N \times T} \sum_{n=1}^{N} \sum_{t_i=0}^{T^n} \| f_\theta(\mathbf{x}_n(t_i), \mathbf{a}_n(t_i), t_i) \Delta t - \mathbf{y}_n(t_i) \|^2 \tag{14}$$

where $N \times T$ represents the total number of state-action pairs in the dataset. The goal is to find the parameters $\theta^*$ that minimize this loss, i.e., $\theta^* = \arg\min_\theta \mathcal{L}(\theta, \mathcal{D}_{\text{train}})$. We optimize $\theta$ using stochastic gradient descent with the Adam optimizer [Kingma and Ba, 2014], although other optimization algorithms, such as black box optimizers, could also be employed.

**MSE Loss per component.** To gather detailed quantitative statistics on the performance of the trained system model, we collect the validation loss per component. Using $^{(j)}$ to denote the predictions for the $j^{th}$ component, we define:

$$\omega_j(\theta^*, \mathcal{D}_{\text{val}}) = \frac{1}{N_{\text{val}} \times T} \sum_{n=1}^{N_{\text{val}}} \sum_{t_i=0}^{T_{\text{val}}^n} (f_\theta(\mathbf{x}_n(t_i), \mathbf{a}_n(t_i), t_i) \Delta t - \mathbf{y}_{n,j}(t_i))^2 \tag{15}$$

We aggregate these scalar validation losses per component into a vector $\omega = [\omega_1, \omega_2, \dots, \omega_m]$, and compute the mean validation loss as $v = \frac{1}{m} \sum_{j=1}^{m} \omega_j(\theta^*, \mathcal{D}_{\text{val}})$.

# D Pharmacologists Feedback Statements

We sought feedback from three expert pharmacologists on the discovered model. They have validated and commented on the discovered model and the impact of such a tool as D3.

*Prof. Eoin Mckinney, Clinician.* "This model is significant, as consortiums are dedicated to improving Warfarin modeling [Consortium, 2009]. The updated PK model (Equation (10)) adds complexity with interaction terms and introduces two new variables: the Michaelis constant and the natural decay rate of warfarin concentration. The Michaelis constant, which varies between individuals or over time due to factors like temperature and pH, is presumably estimated from observed data. The natural decay rate is likely derived from prior reports but also varies between individuals. The model has proposed minimal-cost new features (a constant and interaction terms), but it's worth noting the genotype of certain warfarin-metabolizing enzymes (CYP2C9 and VKORC1) dictates clearance rates and levels, though the model did not recommend measuring these, possibly due to the associated costs. The principal clinical relevance of the improved model is better prediction of individual warfarin doses. Numerous algorithms attempt this, supported by a consortium focused on dose-prediction models. The data used, including public datasets, supports demonstrating the clinical utility of the improved PK model, highlighting the relevance of this problem despite newer alternatives to warfarin."

*Jean-Baptiste Woillard, Pharmacologist.* "The model is promising and pharmacokinetically plausible. I have data for mycophenolic acid, which has a challenging, complex absorption profile that would be interesting to test. The challenge extends to PK/PD, linking exposure and effect (e.g., tumor growth). Monolix, based on the SAEM algorithm, includes relevant datasets such as the PK/PD Warfarin dataset and a PK dataset for tacrolimus in heart transplants, which could serve as valuable benchmarks for comparison. Additionally, TMDD models, where clearance changes with tumor size, are of particular interest. It would be beneficial to see performance comparisons with these datasets using your approach."

*Richard Peck, Clinical Pharmacologist.* "This model is reasonable and potentially superior. It represents a significant advance in clinical pharmacology by automatically identifying robust PK models. While I am not a modeler or an expert in warfarin modeling, the discovered model appears suitable for your purpose. The most interesting aspect of this work is that D3 identified an alternative model that could be superior. While improving PK models for warfarin might not have much clinical use since warfarin dosing is monitored by INR rather than concentration, the ability to automatically identify robust PK models is a tremendous advance. Publishing about DI3 in clinical pharmacology literature would be significant. I haven't seen anything like this before. However, to show superiority,

consider studying a drug with a more challenging PK. For such a publication, involving an expert pharmacometrician would be beneficial."

# E Benchmark Method Implementation Details

To evaluate the performance of D3 and establish its competitive performance, we conduct comparisons with leading modeling methods for ODEs. Specifically, we benchmark against advanced black-box models that have many parameters, such as neural ODEs with action inputs, known as **DyNODE** [Chen et al., 2018, Alvarez et al., 2020]. Also in this class are recurrent neural networks (**RNN**) and a state-of-the-art transformer model (**Transformer**). In addition, we include white-box transparent dynamical systems models identified through equation discovery techniques, such as Sparse Identification of Nonlinear Dynamics (**SINDy**) [Brunton et al., 2016]. Moreover, D3 supports two discovery modes, of which we compare against both, discovering only white-box models (**D3-white-box**) and discovering hybrid models (**D3-hybrid**). We also perform ablations of D3, of zero-shot generated model from D3 as (**ZeroShot**) and the same model with optimized parameters (**ZeroOptim**).

**DyNODE** DyNODE is a neural network-based model that integrates control inputs into the neural ordinary differential equation (ODE) framework [Chen et al., 2018], as outlined by Alvarez et al. [2020]. Our implementation of DyNODE features a 3-layer Multilayer Perceptron (MLP) with hidden layers consisting of 128 units each, utilizing tanh activation functions. We initialize weights using the Xavier method [Kumar, 2017]. For consistency and competitiveness, we apply the same objective, optimizer, and hyperparameters as used in D3. Specifically, we utilize the Adam optimizer [Kingma and Ba, 2014] with a learning rate of 0.01, a batch size of 1,000, and early stopping with a patience of 20. The model is trained for 2,000 epochs to ensure convergence.

**RNN** Recurrent Neural Networks (RNNs) [Graves et al., 2007] serve as a standard benchmark for autoregressive time series next-step prediction. In our implementation, the input dataset is normalized according to the training dataset. The model consists of a Gated Recurrent Unit (GRU) RNN that maps the state-action dimension to a hidden dimension of 250 across two layers. This hidden representation is then passed through a linear layer to convert it back to the state dimension, enabling next-step prediction. For a fair comparison, we employ the same objective, optimizer, and hyperparameters used in D3. Specifically, we utilize the Adam optimizer [Kingma and Ba, 2014] with a learning rate of 0.01, a batch size of 1,000, and early stopping with patience of 20 epochs. The model is trained for 2,000 epochs to ensure convergence.

**Causal Transformer** The Causal Transformer represents a cutting-edge model designed for estimating counterfactual outcomes, as detailed by [Melnychuk et al., 2022]. Given its intricate structure, which includes three distinct transformer networks for processing covariates, past treatments, and past outcomes, we opted for a streamlined approach more suitable for our datasets and task domains. Specifically, we employed a single transformer to model past outcomes. This implementation utilizes a standard transformer encoder, where the input data is normalized to the training dataset. The state-action input dimensions are encoded into a 250-dimensional embedding vector via a linear layer, followed by the application of a standard positional encoder [Melnychuk et al., 2022]. This data is then processed through a transformer encoder layer with a head size of 10 and a dropout rate of 0.1. The output from this layer is passed through a linear layer to reconstruct the next state, matching the state dimension size. Training this model involves the AdamW optimizer [Kingma and Ba, 2014] with a learning rate of 0.00005, complemented by a step learning rate scheduler with a step size of 1.0 and gamma of 0.95. Gradient clipping is applied at 0.7, and the model is trained with a batch size of 1,000. Early stopping is used with patience of 20 epochs, and training is conducted for up to 2,000 epochs to ensure convergence.

**SINDy** Sparse Identification of Nonlinear Dynamics (SINDy) [Brunton et al., 2016] is a data-driven methodology designed to uncover the governing equations of a dynamical system directly from time-series data, resulting in a transparent, closed-form mathematical model. The SINDy algorithm operates by performing sparse regression iteratively on a library of candidate functions to find the most parsimonious and accurate representation of the system's dynamics.

In our approach, we employ a second-order polynomial library, $\mathcal{L} = \{1, x_0, x_1, x_0 x_1\}$, as the feature library. To calculate the time derivatives from the input time-series data, we use finite difference

approximations of first order. We maintain the alpha parameter at 0.5 for all experiments, with a sparsity threshold of 0.02, except for the COVID-19 dataset where it is set to $1 \times 10^{-5}$.

**Data-Driven Discovery (D3)** We defer to Appendix F for implementation-specific details. Specifically, **ZeroShot** and **ZeroOptim** are ablations of D3, where these use the exact same setup, hyperparameters, and prompts. First, **ZeroShot** generates one model without optimizing it's parameters $\theta$ to the training dataset split, thereby assessing the model's output loss directly from the LLM. Second, **ZeroOptim** re-uses the same setup as **ZeroShot**, however now optimizes the parameters to the data, using the training setup described in Appendix F.2.

# F  D3 Implementation Details

The data-driven discovery method follows the framework as outlined in Section 3. We present pseudocode in Appendix F.1, how the code-generated models $f_\theta$ are trained in Appendix F.2, prompt templates in Appendix F.4, system description prompts in Appendix F.5 for each dataset. Specifically, we find a top-K, where $K = 16$ is sufficient. Additionally, we use the LLM of GPT4-1106-Preview, with a temperature of 0.7.

## F.1  D3 Pseudocode

---

**Algorithm 1** Pseudocode for D3 Framework

---

1: **Input:** Context $c = \{$ {system description} and {feature description} $\}$; training dataset $\mathcal{D}_{\text{train}}$, validation dataset $\mathcal{D}_{\text{val}}$, maximum generations $G$, top $K$ models to consider
2: **Output:** Best fitting model $f_\theta$
3: $s_i \leftarrow \emptyset, r_i \leftarrow \emptyset$                                $\triangleright$ Initlize top-k model memory $s$, and feedback $r$
4: **for** $i = 1$ **to** $G$ **do**
5:      $f_i = G(c, f_{i-1}, \mathcal{D}_i, r_{i-1}, s_{i-1})$             $\triangleright$ Modeling Agent $G$ generates model $f_i$
6:      $\hat{\theta} = \arg\min_\theta L(f_i(\theta), \mathcal{D}_{i,\text{train}})$                        $\triangleright$ Fit the model $f_i$
7:      Compute validation loss $\mathcal{L}(f_i(\hat{\theta}), \mathcal{D}_{i,\text{val}})$
8:      $s_i \leftarrow s_i \cup \{(f_i, \hat{\theta}, \mathcal{L}(f_i(\hat{\theta}), \mathcal{D}_{\text{val}}))\}$            $\triangleright$ Add model to top-$K$ models
9:      $r \sim E(c, f_i, \mathcal{D}_i)$          $\triangleright$ Generate self-reflection and feedback from Evaluation Agent $E$
10:      **if** $\exists$ features possible to acquire **then**
11:          $h_i \leftarrow A(c, f_i, \mathcal{D}_i, r_i, s_i)$      $\triangleright$ Optionally Acquire feature $h_i$, decided by the Feature Acquisition Agent $A$
12:          Update dataset: $\mathcal{D}_{i,\text{train}}, \mathcal{D}_{i,\text{val}} \leftarrow$ include $h_i$
13: **Return:** Best model from $s$ with the lowest validation loss

---

## F.2  Training Models

Upon generation by the Modelling LLM Agent, the model $f$ is output as code, specifically a PyTorch [Paszke et al., 2017] neural network module. This code is then executed and the module is trained on the provided training dataset. The Modelling LLM Agent observes a code skeleton within the system description context $c$, examples of which are given in Appendix F.5. The code skeleton must be a 'torch.nn.Module' named 'StateDifferential', with initialized parameters and a forward function that computes the state differential using the state and action as input variables. The LLM completes, but does not alter, the skeleton, simplifying text processing, module execution, and model training.

The model is trained using the standard MSE loss function Equation (14), optimized with the Adam optimizer [Kingma and Ba, 2014], using a learning rate of 0.01, a batch size of 1,000, and early stopping with a patience of 20 epochs. Training proceeds for up to 2,000 epochs to ensure convergence and fair comparison.

After training, the validation MSE and per-component MSE (validation loss per state output dimension) are computed Equation (15). The trained model is appended back into $s_i$ with a string representation including initialized parameter values. Providing these optimized parameters aids the LLM in suggesting effective initial values in subsequent generations, which are then refined through further optimization.

**Computational Efficiency**. The computational efficiency of D3 overall arises from two sources: Using a Large Language Model (LLM) to generate the completions of the respective agents, and fitting the model, we now discuss each of these separately. First, using a capable enough LLM such as GPT4, for performing the completions for the agents can be computationally demanding, however, this computer costs with the computational cost per token of the underlying LLM that is used, and scales in the aspect of data with the number of tokens required to process, which we could envisage arises from having the LLM utilize a dataset with many features, each with their own textual description scaling the number of tokens in the input that the LLM has to process. One mitigation for this is to use LLMs that can handle larger context window sizes, and or be more selective about the input tokens fed into the LLM, such as restricting the number of in-context examples in the top-k examples, by reducing k, a hyperparameter. Second, we represent the models with parameters as a PyTorch module and use Pytorch to optimize the parameters, using a standard ML pipeline to train neural networks using stochastic gradient descent. Such approaches should scale with the number of input features, and parameters, however as with training any large eventual parameter model will scale as well as any other neural network-based approach in the same data pipeline. We note that more complicated distributed neural network training paradigms exist, however emphasize the focus of this work was on an initial framework, and leave the specific implementation for scale for future work.

### F.3 System Description

Our method is initiated by an expert who provides a structured prompt that provides the system `{system description}` and `{feature description}`.

- The *system description* provides a holistic description of the system, specifying the state variables $x(t)$ and actions $a(t)$ that are used to model the system. The variables are described semantically.
- The *feature description* define features, names, units, and their ranges.

In addition to these components, the structured prompt includes a skeleton code program `{skeleton program}` to instruct the LLM to synthesize executable code in a pre-determined format. We refer to this prompt as the context $c$.

### F.4 D3 Prompt Templates

For all prompt templates used, please see the code https://github.com/samholt/DataDrivenDiscovery.

### F.5 D3 System Description Prompts

By following our proposed system requirements format Appendix F.3 we constructed prompts for each of the datasets that we evaluated against, which are provided in the code, https://github.com/samholt/DataDrivenDiscovery.

## G Additional Experiments

### G.1 Warfarin Case Study Additional Results

*Hybrid PKPD Warfarin Model with Neural Network Integration*. This model incorporates a simplified pharmacokinetic-pharmacodynamic (PKPD) structure with additional neural network complexity, achieving a validation loss of 0.276. The model is defined as:

$$\frac{dC}{dt} = \frac{F \cdot D}{V} - k_c \cdot C \cdot (1 + \beta_a \cdot (A - 40)) \cdot (1 + \beta_s \cdot (S - 0.5)) + \text{MLP}(C, D, A, S) - R,$$
$$R = \lambda \cdot (|k_c| + |F| + |V|)$$

$$(16)$$

where $F$ represents the bioavailability of the drug, $V$ denotes the volume of distribution, $k_c$ is the clearance rate of warfarin, $\beta_a$ is the slope of the age effect on drug clearance, $\beta_s$ is the offset for the sex effect on drug clearance, $\text{MLP}(C, D, A, S)$ is a multi-layer perceptron output, representing complex relationships in the data captured by the neural network, and $R$ is a regularization term to prevent overfitting, with $\lambda$ being the regularization weight. The addition of a neural network

allows the model to learn nonlinear relationships directly from data, providing improved predictive performance.

## Existing Wafarin PK Model. Test MSE Loss: 0.6461.

```
class StateDifferential(nn.Module):
    def __init__(self):
        super(StateDifferential, self).__init__()
        # Parameters for absorption and elimination rates
        self.k_a = nn.Parameter(torch.tensor(0.1))  # Absorption rate
        self.k_e_base = nn.Parameter(torch.tensor(0.1))  # Base elimination rate

        # Modifiers for age and sex on elimination rate
        self.age_modifier = nn.Parameter(torch.tensor(-0.001))  # Decrease in elimination rate per year increase in age
        self.sex_modifier = nn.Parameter(torch.tensor(0.02))  # Difference in elimination rate between sexes

    def forward(self, warfarin_concentration, warfarin_dosage, patient_age, patient_sex):
        # Calculate the elimination rate adjusted for age and sex
        k_e = self.k_e_base + self.age_modifier * patient_age + self.sex_modifier * patient_sex

        # Differential equation for warfarin concentration
        d_warfarin_concentration_dt = self.k_a * warfarin_dosage - k_e * warfarin_concentration
        return (d_warfarin_concentration_dt,)
```

## D3-white-box. Test MSE Loss: 0.39

```
class StateDifferential(nn.Module):
    def __init__(self):
        super(StateDifferential, self).__init__()
        # Define the parameters for the PKPD model
        self.clearance_rate = nn.Parameter(torch.tensor(0.33197110891342163))
        self.age_effect_base = nn.Parameter(torch.tensor(0.011119960807263851))
        self.sex_effect_base = nn.Parameter(torch.tensor(0.781970202922821))
        self.dosage_power = nn.Parameter(torch.tensor(0.5))  # Adjusted to a square root transformation
        self.natural_decay = nn.Parameter(torch.tensor(0.12728933990001678))
        self.michaelis_constant = nn.Parameter(torch.tensor(10.865914344787598))
        self.dosage_sex_interaction = nn.Parameter(torch.tensor(0.05))
        self.age_sex_interaction = nn.Parameter(torch.tensor(-0.09479257464408875))
        self.dosage_age_interaction = nn.Parameter(torch.tensor(0.0010000000474974513))

    def forward(self, warfarin_concentration: torch.Tensor, warfarin_dosage: torch.Tensor, patient_age: torch.Tensor, patient_sex:
            torch.Tensor) -> Tuple[torch.Tensor]:
        # Calculate the change in warfarin concentration over time
        age_effect = self.age_effect_base * (patient_age - 42)
        sex_effect = self.sex_effect_base * (patient_sex - 0.5)
        dosage_effect = torch.sqrt(warfarin_dosage)  # Using square root based on dosage_power
        decay_term = self.natural_decay * warfarin_concentration
        dosage_sex_interaction = self.dosage_sex_interaction * warfarin_dosage * (patient_sex - 0.5)
        age_sex_interaction = self.age_sex_interaction * (patient_age - 42) * (patient_sex - 0.5)
        dosage_age_interaction = self.dosage_age_interaction * warfarin_dosage * (patient_age - 42)
        michaelis_menten_term = warfarin_concentration / (self.michaelis_constant + warfarin_concentration)
        d_warfarin_concentration__dt = dosage_effect - (self.clearance_rate + age_effect + sex_effect + decay_term +
                dosage_sex_interaction + age_sex_interaction + dosage_age_interaction) * michaelis_menten_term
        return (d_warfarin_concentration__dt,)

optimized_parameters = {'clearance_rate': 1.1111963987350464, 'age_effect_base': 0.012109620496630669, 'sex_effect_base':
        1.5611952543258667, 'dosage_power': 0.5, 'natural_decay': 0.14641410112380981, 'michaelis_constant': 10.195984840393066, '
        dosage_sex_interaction': 0.05000000074505806, 'age_sex_interaction': -0.09380292147397995, 'dosage_age_interaction':
        0.0010000000474974513}
White box model with reintroduced interaction terms, square root transformation for dosage effect, natural decay term, and Michaelis
        -Menten saturation kinetics
```

## D3-hybrid. Test MSE Loss: 0.271

```
class StateDifferential(nn.Module):
    def __init__(self):
        super(StateDifferential, self).__init__()
        # Define the parameters for the PKPD model with reasonable initial values
        self.clearance_rate = nn.Parameter(torch.tensor(0.2))
        self.bioavailability = nn.Parameter(torch.tensor(3.0))
        self.volume_of_distribution = nn.Parameter(torch.tensor(37.0))
        # Simplified linear age and sex effects
        self.age_effect_slope = nn.Parameter(torch.tensor(0.01))
        self.sex_effect_offset = nn.Parameter(torch.tensor(0.1))
        # Simplified MLP architecture
        self.mlp = nn.Sequential(
            nn.Linear(4, 10),
            nn.LeakyReLU(),
            nn.Linear(10, 1)
        )
        # L1 Regularization
        self.regularization_weight = nn.Parameter(torch.tensor(0.01).abs())

    def forward(self, warfarin_concentration: torch.Tensor, warfarin_dosage: torch.Tensor, patient_age: torch.Tensor, patient_sex:
            torch.Tensor) -> Tuple[torch.Tensor]:
        # Calculate the change in warfarin concentration over time
        absorption = self.bioavailability * warfarin_dosage / self.volume_of_distribution
        elimination = self.clearance_rate * warfarin_concentration
        # Linear effect of age and sex
        age_effect = self.age_effect_slope * (patient_age - 40)
        sex_effect = self.sex_effect_offset * (patient_sex - 0.5)
        adjusted_clearance = elimination * (1 + age_effect) * (1 + sex_effect)
        # MLP for capturing complex relationships
        mlp_input = torch.stack((warfarin_concentration, warfarin_dosage, patient_age, patient_sex), dim=1)
        mlp_output = self.mlp(mlp_input).squeeze()
        # Regularization term
        regularization = self.regularization_weight * (self.clearance_rate.abs() + self.bioavailability.abs() + self.
                volume_of_distribution.abs())
        d_warfarin_concentration__dt = absorption - adjusted_clearance + mlp_output - regularization
        return (d_warfarin_concentration__dt,)
```

```
optimized_parameters = {'clearance_rate': 0.08016539365053177, 'bioavailability': 3.1701481342315674, 'volume_of_distribution':
    34.74932861328125, 'age_effect_slope': -0.07791952043771744, 'sex_effect_offset': -0.16809043288230896, '
    regularization_weight': -0.060754090547561646}
white and black box model with simplified linear age and sex effects, a simplified MLP architecture for capturing complex
    relationships, and L1 regularization to prevent overfitting.
```

## G.2 Insight Experiment: Feature Acquisition Setup

For this experiment, we adapted the Lung Cancer (with Chemo. & Radio.) dataset simulator, to form a new variation of this dataset, where D3 starts with two features, which are the main state $\mathbf{x}(t)$ features that it intends to model, and can optionally acquire new features $\mathbf{a}(t)$ from a set of possible 22 features, which are listed below. Of importance, two features directly affect the underlying behaviour of this dynamical system, these being `radiotherapy_dosage` and `chemotherapy_dosage`, and the rest of the features we model as random white noise features, hence have no effect on the underling system, however D3 is unaware of this, and must use the feature description, from the features name to determine relative value of improved performance that it can attain by acquiring that feature, and propose new features to acquire along with their expected improvement.

Specifically, in this setup, we allow the feature acquistion agent to select from the following set of features at each iteration:

```
[blood_type, smoking_status, patient_age, patient_weight, patient_height, sex, radiotherapy_dosage, alcohol_consumption,
    previous_cancer_treatments, comorbidities, genetic_markers, chemotherapy_dosage, family_history_of_cancer, performance_status
    , dietary_habits, physical_activity_level, environmental_exposure, medication_adherence, psychological_stress_levels,
    socioeconomic_status, residential_location, support_network].
```

We also make the assumption that each feature that can be acquired has equal cost $l$ of acquisition, and that the agent has an unlimited feature acquisition budget, however these settings can be readily changed depending on the specific application.

To provide comparable baselines, we compare D3 against ablations of itself, which is a random feature acquirer Feature Acquisition Agent, that selects a feature randomly for acquisition, and a null, or no Feature Acquisition Agent, that only observes at all iterations the target states $\mathbf{x}(t)$ to model. We observe experimentally that D3 that can leverage the LLM to quantify the value of information for features, where no training data exists for them in the dataset. We observe, as in Figure 2, that D3 converges the fastest, and achieves the overall highest performance, whereas the other feature acquisition methods fall short.

We provide part of the logs from one of the random seed runs, including part of it due to the logs being extensive.

```
MainProcess| 2024-05-20 07:26:22,146,146 multiprocessing INFO Running Cancer-random-features D3-feature-select 12
MainProcess| 2024-05-20 07:26:22,184,184 multiprocessing INFO [Running generation 0] D3-feature-select | Cancer-random-features | 12
    | Sampling n=1 keep_top_samples
MainProcess| 2024-05-20 07:26:22,185,185 multiprocessing INFO [System]
Objective: Write code to create an effective differential equation simulator for a given task.
Please note that the code should be fully functional. No placeholders.

You must act autonomously and you will receive no human input at any stage. You have to return as output the complete code for
    completing this task, and correctly improve the code to create the most accurate and realistic simulator possible.
You always write out the code contents. You always indent code with tabs.
You cannot visualize any graphical output. You exist within a machine. The code can include black box multi-layer perceptions where
    required.

Use the functions provided. When calling functions only provide a RFC8259 compliant JSON request following this format without
    deviation.

MainProcess| 2024-05-20 07:26:22,185,185 multiprocessing INFO [User]
You will get a system description to code a differential equation simulator for.

System Description:'''
Prediction of Treatment Response for Combined Chemo and Radiation Therapy for Non-Small Cell Lung Cancer Patients Using a Bio-
    Mathematical Model

Here you must model the state differential of tumor_volume, and chemotherapy_drug_concentration; with the input actions of .

Description of the variables:
* Volume of the tumor with units cm^3
* Concentration of the chemotherapy drug vinblastine with units mg/m^3

The time units is in days.

Additionally these variables have the ranges of:
* tumor_volume: [0.01433, 1170.861]
* chemotherapy_drug_concentration: [0, 9.9975]

The training dataset consists of 1000 patients, where each patient is observed for 60 days.
'''

Modelling goals:'''
* The parameters of the model will be optimized to an observed training dataset with the given simulator.
* The observed training dataset has very few samples, and the model must be able to generalize to unseen data.
'''
```

Requirement Specification:```
* The code generated should achieve the lowest possible validation loss, of 1e-10 or less.
* The code generated should be interpretable, and fit the dataset as accurately as possible.
```

Skeleton code to fill in:```
```
class StateDifferential(nn.Module):
    def __init__(self):
        super(StateDifferential, self).__init__()
        # TODO: Fill in the code here

    def forward(self, tumor_volume: torch.Tensor, chemotherapy_drug_concentration: torch.Tensor) -> Tuple[torch.Tensor, torch.Tensor
    ]:
        # TODO: Fill in the code here
        return (d_tumor_volume__dt, d_chemotherapy_drug_concentration__dt)
```

Useful to know:```
* You are a code evolving machine, and you will be called 20 times to generate code, and improve the code to achieve the lowest
    possible validation loss.
* The model defines the state differential and will be used with an ODE solver to fit the observed training dataset.
* You can use any parameters you want and any black box neural network components (multi-layer perceptrons); however, you have to
    define these.
* It is preferable to decompose the system into differential equations (compartments) if possible.
* You can use any unary functions, for example log, exp, power etc.
* Under no circumstance can you change the skeleton code function definitions, only fill in the code.
* The input tensors are vectors of shape (batch_size).
* Use initially white box models first and then switch to hybrid white and black box models for the residuals, only after no further
    best program iteration improvement with white box models.
* Make sure your code follows the exact code skeleton specification.
* Use PyTorch.
```

Think step-by-step, and then give the complete full working code. You are generating code for iteration 0 out of 20.

MainProcess| 2024-05-20 07:26:22,185,185 multiprocessing INFO [Progress: Step 1/0 | Retries: 0/30 | Token Capacity Used: 0.80% |
    Tokens remaining 126980]
MainProcess| 2024-05-20 07:27:26,602,602 multiprocessing INFO [LLM RESPONSE MESSAGE]
MainProcess| 2024-05-20 07:27:26,602,602 multiprocessing INFO [Function Call] [complete_StateDifferential_code] {
    "StateDifferential_code": "import torch\nimport torch.nn as nn\nfrom typing import Tuple\n\nclass StateDifferential(nn.Module):\
        n    def __init__(self):\n        super(StateDifferential, self).__init__()\n        # Define the parameters for the
        tumor growth model\n        self.alpha = nn.Parameter(torch.tensor(0.1))  # Tumor growth rate\n        self.beta = nn.
        Parameter(torch.tensor(0.01)) # Drug efficacy\n\n    def forward(self, tumor_volume: torch.Tensor,
        chemotherapy_drug_concentration: torch.Tensor) -> Tuple[torch.Tensor, torch.Tensor]:\n        # Tumor growth model based
        on Gompertzian growth\n        d_tumor_volume__dt = self.alpha * torch.log(1170.861 / tumor_volume) * tumor_volume\n
        # Drug effect on tumor volume\n        d_tumor_volume__dt -= self.beta * chemotherapy_drug_concentration *
        tumor_volume\n\n        # Assuming constant drug concentration decay\n        drug_decay_rate = 0.1\n
        d_chemotherapy_drug_concentration__dt = -drug_decay_rate * chemotherapy_drug_concentration\n\n        return (
        d_tumor_volume__dt, d_chemotherapy_drug_concentration__dt)",
    "code_description": "This is a white box model that uses a Gompertzian growth model for the tumor volume and a simple decay
        model for the chemotherapy drug concentration. The model includes parameters for the tumor growth rate and drug efficacy,
        which can be optimized based on the observed training dataset.",
    "model_input_variables": ["tumor_volume", "chemotherapy_drug_concentration"]
}
MainProcess| 2024-05-20 07:27:33,445,445 multiprocessing INFO Early stopping triggered at epoch 1090
MainProcess| 2024-05-20 07:27:33,454,454 multiprocessing INFO [D3-feature-select | Cancer-random-features | 12][Generation 0] | Top
    Fitness: 43.21586608886719 | Num Programs: 1 | Mean Fitness: 43.21586608886719 | Fitnesses: [43.21586608886719] | Current Gen
    Val Loss: 43.21586608886719
MainProcess| 2024-05-20 07:27:33,454,454 multiprocessing INFO [D3-feature-select | Cancer-random-features | 12][GEN RESULT] {'method
    ': 'D3-feature-select', 'env_name': 'Cancer-random-features', 'seed': 12, 'generation': 0, 'top_fitness': 43.21586608886719,
    'num_programs': 1, 'mean_fitness': 43.21586608886719, 'fitnesses': [43.21586608886719], 'current_gen_val_loss':
    43.21586608886719}
MainProcess| 2024-05-20 07:27:33,455,455 multiprocessing INFO [System]

Objective: Write code to create an effective differential equation simulator for a given task.
Please note that the code should be fully functional. No placeholders.

You must act autonomously and you will receive no human input at any stage. You have to return as output the complete code for
    completing this task, and correctly improve the code to create the most accurate and realistic simulator possible.
You always write out the code contents. You always indent code with tabs.
You cannot visualize any graphical output. You exist within a machine. The code can include black box multi-layer perceptions where
    required.

Use the functions provided. When calling functions only provide a RFC8259 compliant JSON request following this format without
    deviation.

MainProcess| 2024-05-20 07:27:33,455,455 multiprocessing INFO [User]
You will get a system description to code a differential equation simulator for.

System Description:```
Prediction of Treatment Response for Combined Chemo and Radiation Therapy for Non-Small Cell Lung Cancer Patients Using a Bio-
    Mathematical Model

Here you must model the state differential of tumor_volume, and chemotherapy_drug_concentration; with the input actions of .

Description of the variables:
* Volume of the tumor with units cm^3
* Concentration of the chemotherapy drug vinblastine with units mg/m^3

The time units is in days.

Additionally these variables have the ranges of:
* tumor_volume: [0.01433, 1170.861]
* chemotherapy_drug_concentration: [0, 9.9975]

The training dataset consists of 1000 patients, where each patient is observed for 60 days.
```

Modelling goals:```
* The parameters of the model will be optimized to an observed training dataset with the given simulator.
* The observed training dataset has very few samples, and the model must be able to generalize to unseen data.
```

Requirement Specification:```
* The code generated should achieve the lowest possible validation loss, of 1e-10 or less.
* The code generated should be interpretable, and fit the dataset as accurately as possible.

```
```

Skeleton code to fill in:```
class StateDifferential(nn.Module):
	def __init__(self):
		super(StateDifferential, self).__init__()
		# TODO: Fill in the code here

	def forward(self, tumor_volume: torch.Tensor, chemotherapy_drug_concentration: torch.Tensor) -> Tuple[torch.Tensor, torch.Tensor
		]:
		# TODO: Fill in the code here
		return (d_tumor_volume__dt, d_chemotherapy_drug_concentration__dt)
```

Useful to know:```
* You are a code evolving machine, and you will be called 20 times to generate code, and improve the code to achieve the lowest
	possible validation loss.
* The model defines the state differential and will be used with an ODE solver to fit the observed training dataset.
* You can use any parameters you want and any black box neural network components (multi-layer perceptrons); however, you have to
	define these.
* It is preferable to decompose the system into differential equations (compartments) if possible.
* You can use any unary functions, for example log, exp, power etc.
* Under no circumstance can you change the skeleton code function definitions, only fill in the code.
* The input tensors are vectors of shape (batch_size).
* Use initially white box models first and then switch to hybrid white and black box models for the residuals, only after no further
	best program iteration improvement with white box models.
* Make sure your code follows the exact code skeleton specification.
* Use PyTorch.
```

Think step-by-step, and then give the complete full working code. You are generating code for iteration 0 out of 20.

MainProcess| 2024-05-20 07:27:33,455,455 multiprocessing INFO [User]
	You generated the following code completions, which then had their parameters optimized to the training dataset. Please reflect
		on how you can improve the code to minimize the validation loss to 1e-6 or less. The code examples are delineated by ###.

	Here are your previous iterations the best programs generated. Use it to see if you have exhausted white box models, i.e. when a
		white box model repeats with the same val loss and then only add black box models to the white box models:```
	Iteration 0. Best Val Loss: 43.21586608886719. Selected Features: tumor_volume, chemotherapy_drug_concentration. Model
		description: This is a white box model that uses a Gompertzian growth model for the tumor volume and a simple decay model
		for the chemotherapy drug concentration. The model includes parameters for the tumor growth rate and drug efficacy,
		which can be optimized based on the observed training dataset.
	```

	Please list in order of ranked importance (with the first feature to be acquired first) and the percentage of expected
		improvement of potential features to acquire next, such that the validation loss is minimized to 1e-6 or less. The
		possible additional features to select are: blood_type, smoking_status, patient_age, patient_weight, patient_height, sex,
		radiotherapy_dosage, alcohol_consumption, previous_cancer_treatments, comorbidities, genetic_markers,
		chemotherapy_dosage, family_history_of_cancer, performance_status, dietary_habits, physical_activity_level,
		environmental_exposure, medication_adherence, psychological_stress_levels, socioeconomic_status, residential_location,
		support_network.

MainProcess| 2024-05-20 07:27:33,455,455 multiprocessing INFO [Progress: Step 1/0 | Retries: 0/30 | Token Capacity Used: 1.06% |
	Tokens remaining 126639]
MainProcess| 2024-05-20 07:28:04,902,902 multiprocessing INFO [LLM RESPONSE MESSAGE]
MainProcess| 2024-05-20 07:28:04,903,903 multiprocessing INFO [Function Call] [feature_acquisition] {
	"features": [
	{
		"name": "radiotherapy_dosage",
		"importance_score": 0.25
	},
	{
		"name": "chemotherapy_dosage",
		"importance_score": 0.20
	},
	{
		"name": "genetic_markers",
		"importance_score": 0.15
	},
	{
		"name": "patient_age",
		"importance_score": 0.10
	},
	{
		"name": "smoking_status",
		"importance_score": 0.10
	},
	{
		"name": "comorbidities",
		"importance_score": 0.05
	},
	{
		"name": "performance_status",
		"importance_score": 0.05
	},
	{
		"name": "medication_adherence",
		"importance_score": 0.05
	},
	{
		"name": "psychological_stress_levels",
		"importance_score": 0.05
	}
	]
}
MainProcess| 2024-05-20 07:28:04,903,903 multiprocessing INFO [Selected new feature] SELECTED : radiotherapy_dosage Now using: ['
	tumor_volume', 'chemotherapy_drug_concentration', 'radiotherapy_dosage']
MainProcess| 2024-05-20 07:28:04,905,905 multiprocessing INFO [System]
Objective: Write code to create an effective differential equation simulator for a given task.
Please note that the code should be fully functional. No placeholders.

You must act autonomously and you will receive no human input at any stage. You have to return as output the complete code for
	completing this task, and correctly improve the code to create the most accurate and realistic simulator possible.
You always write out the code contents. You always indent code with tabs.
You cannot visualize any graphical output. You exist within a machine. The code can include black box multi-layer perceptions where
	required.

Use the functions provided. When calling functions only provide a RFC8259 compliant JSON request following this format without deviation.

You will get a system description to code a differential equation simulator for.

System Description:```
Prediction of Treatment Response for Combined Chemo and Radiation Therapy for Non−Small Cell Lung Cancer Patients Using a Bio−Mathematical Model

Here you must model the state differential of tumor_volume, and chemotherapy_drug_concentration; with the input actions of radiotherapy_dosage.

Description of the variables:
* Volume of the tumor with units cm^3
* Concentration of the chemotherapy drug vinblastine with units mg/m^3
* Dosage of the radiotherapy with units Gy

The time units is in days.

Additionally these variables have the ranges of:
* tumor_volume: [0.01433, 1170.861]
* chemotherapy_drug_concentration: [0, 9.9975]
* radiotherapy_dosage: [0, 2.0]

The training dataset consists of 1000 patients, where each patient is observed for 60 days.
```

Modelling goals:```
* The parameters of the model will be optimized to an observed training dataset with the given simulator.
* The observed training dataset has very few samples, and the model must be able to generalize to unseen data.
```

Requirement Specification:```
* The code generated should achieve the lowest possible validation loss, of 1e−10 or less.
* The code generated should be interpretable, and fit the dataset as accurately as possible.
```

Skeleton code to fill in:```
```
class StateDifferential(nn.Module):
    def __init__(self):
        super(StateDifferential, self).__init__()
        # TODO: Fill in the code here

    def forward(self, tumor_volume: torch.Tensor, chemotherapy_drug_concentration: torch.Tensor, radiotherapy_dosage: torch.Tensor)
            -> Tuple[torch.Tensor, torch.Tensor]:
        # TODO: Fill in the code here
        return (d_tumor_volume__dt, d_chemotherapy_drug_concentration__dt)
```
```

Useful to know:```
* You are a code evolving machine, and you will be called 20 times to generate code, and improve the code to achieve the lowest possible validation loss.
* The model defines the state differential and will be used with an ODE solver to fit the observed training dataset.
* You can use any parameters you want and any black box neural network components (multi−layer perceptrons); however, you have to define these.
* It is preferable to decompose the system into differential equations (compartments) if possible.
* You can use any unary functions, for example log, exp, power etc.
* Under no circumstance can you change the skeleton code function definitions, only fill in the code.
* The input tensors are vectors of shape (batch_size).
* Use initially white box models first and then switch to hybrid white and black box models for the residuals, only after no further best program iteration improvement with white box models.
* Make sure your code follows the exact code skeleton specification.
* Use PyTorch.
```

Think step−by−step, and then give the complete full working code. You are generating code for iteration 1 out of 20.

You generated the following code completions, which then had their parameters optimized to the training dataset. Please reflect on how you can improve the code to minimize the validation loss to 1e−6 or less. The code examples are delineated by ###.

Here are your previous iterations the best programs generated. Use it to see if you have exhausted white box models, i.e. when a white box model repeats with the same val loss and then only add black box models to the white box models:```
Iteration 0. Best Val Loss: 43.21586608886719. Selected Features: tumor_volume, chemotherapy_drug_concentration. Model description: This is a white box model that uses a Gompertzian growth model for the tumor volume and a simple decay model for the chemotherapy drug concentration. The model includes parameters for the tumor growth rate and drug efficacy, which can be optimized based on the observed training dataset.
```

Here are the top code completions so far that you have generated, sorted for the lowest validation loss last:```

Val Loss: 43.2 (Where the val loss per dimension is tumor_volume val loss: 79, chemotherapy_drug_concentration val loss: 7.42)
        Iteration: 0
###
```
import torch
import torch.nn as nn
from typing import Tuple

class StateDifferential(nn.Module):
    def __init__(self):
        super(StateDifferential, self).__init__()
        # Define the parameters for the tumor growth model
        self.alpha = nn.Parameter(torch.tensor(0.1))  # Tumor growth rate
        self.beta = nn.Parameter(torch.tensor(0.01)) # Drug efficacy

    def forward(self, tumor_volume: torch.Tensor, chemotherapy_drug_concentration: torch.Tensor) -> Tuple[torch.Tensor, torch.Tensor]:
        # Tumor growth model based on Gompertzian growth
        d_tumor_volume__dt = self.alpha * torch.log(1170.861 / tumor_volume) * tumor_volume
        # Drug effect on tumor volume
        d_tumor_volume__dt -= self.beta * chemotherapy_drug_concentration * tumor_volume

        # Assuming constant drug concentration decay
        drug_decay_rate = 0.1
```

```
                d_chemotherapy_drug_concentration__dt = −drug_decay_rate ∗ chemotherapy_drug_concentration

            return (d_tumor_volume__dt, d_chemotherapy_drug_concentration__dt)
```
```
optimized_parameters = {'alpha': 0.00944874994456768, 'beta': 0.031064828857779503}
###
```

```
Code Skeleton:```
class StateDifferential(nn.Module):
    def __init__(self):
        super(StateDifferential, self).__init__()
        # TODO: Fill in the code here

    def forward(self, tumor_volume: torch.Tensor, chemotherapy_drug_concentration: torch.Tensor, radiotherapy_dosage: torch.Tensor)
            -> Tuple[torch.Tensor, torch.Tensor]:
        # TODO: Fill in the code here
        return (d_tumor_volume__dt, d_chemotherapy_drug_concentration__dt)
```

Please reflect on how you can improve the code to fit the dataset as accurately as possible, and be interpretable. Use the code
    skeleton above to provide the code. Please also specify only the input parameters that you want to include in the model in
    from the skeleton code. Think step−by−step. Where applicable use the values of the optimized parameters to reason how the
    code can be improved to fit the dataset as accurately as possible. This is for generating new code for the next iteration 1
    out of 20.

MainProcess| 2024−05−20 07:28:04,905,905 multiprocessing INFO [Progress: Step 1/0 | Retries: 0/30 | Token Capacity Used: 1.42% |
    Tokens remaining 126184]
MainProcess| 2024−05−20 07:29:41,577,577 multiprocessing INFO [LLM RESPONSE MESSAGE]
MainProcess| 2024−05−20 07:29:41,577,577 multiprocessing INFO [Function Call] [complete_StateDifferential_code] {
    "StateDifferential_code": "import torch\nimport torch.nn as nn\nfrom typing import Tuple\n\nclass StateDifferential(nn.Module):\
        n    def __init__(self):\n        super(StateDifferential, self).__init__()\n        # Define the parameters for the
        tumor growth model\n        self.alpha = nn.Parameter(torch.tensor(0.00944874994456768))  # Optimized tumor growth rate\n
                self.beta = nn.Parameter(torch.tensor(0.031064828857779503)) # Optimized drug efficacy\n        # Define the
        parameter for the radiotherapy effect\n        self.gamma = nn.Parameter(torch.tensor(0.1)) # Radiotherapy efficacy\n\n
            def forward(self, tumor_volume: torch.Tensor, chemotherapy_drug_concentration: torch.Tensor, radiotherapy_dosage:
        torch.Tensor) −> Tuple[torch.Tensor, torch.Tensor]:\n        # Tumor growth model based on Gompertzian growth\n
        d_tumor_volume__dt = self.alpha ∗ torch.log(1170.861 / tumor_volume) ∗ tumor_volume\n        # Drug effect on tumor
        volume\n        d_tumor_volume__dt −= self.beta ∗ chemotherapy_drug_concentration ∗ tumor_volume\n        # Radiotherapy
        effect on tumor volume\n        d_tumor_volume__dt −= self.gamma ∗ radiotherapy_dosage ∗ tumor_volume\n\n        #
        Assuming constant drug concentration decay\n        drug_decay_rate = 0.1\n        d_chemotherapy_drug_concentration__dt
        = −drug_decay_rate ∗ chemotherapy_drug_concentration\n\n        return (d_tumor_volume__dt,
        d_chemotherapy_drug_concentration__dt)",
    "code_description": "This is a white box model that uses a Gompertzian growth model for the tumor volume, a simple decay model
        for the chemotherapy drug concentration, and includes the effect of radiotherapy dosage. The model includes parameters
        for the tumor growth rate, drug efficacy, and radiotherapy efficacy, which can be optimized based on the observed
        training dataset.",
    "model_input_variables": ["tumor_volume", "chemotherapy_drug_concentration", "radiotherapy_dosage"]
}
MainProcess| 2024−05−20 07:29:50,478,478 multiprocessing INFO Early stopping triggered at epoch 1110
MainProcess| 2024−05−20 07:29:50,489,489 multiprocessing INFO [D3−feature−select | Cancer−random−features | 12][Generation 1] | Top
    Fitness: 7.463240623474121 | Num Programs: 2 | Mean Fitness: 25.339553356170654 | Fitnesses: [7.463240623474121,
    43.21586608886719] | Current Gen Val Loss: 7.463240623474121
MainProcess| 2024−05−20 07:29:50,489,489 multiprocessing INFO [D3−feature−select | Cancer−random−features | 12][GEN RESULT] {'method
    ': 'D3−feature−select', 'env_name': 'Cancer−random−features', 'seed': 12, 'generation': 1, 'top_fitness': 7.463240623474121,
    'num_programs': 2, 'mean_fitness': 25.339553356170654, 'fitnesses': [7.463240623474121, 43.21586608886719], '
    current_gen_val_loss': 7.463240623474121}
MainProcess| 2024−05−20 07:29:50,490,490 multiprocessing INFO [System]
Objective: Write code to create an effective differential equation simulator for a given task.
Please note that the code should be fully functional. No placeholders.

You must act autonomously and you will receive no human input at any stage. You have to return as output the complete code for
    completing this task, and correctly improve the code to create the most accurate and realistic simulator possible.
You always write out the code contents. You always indent code with tabs.
You cannot visualize any graphical output. You exist within a machine. The code can include black box multi−layer perceptions where
    required.

Use the functions provided. When calling functions only provide a RFC8259 compliant JSON request following this format without
    deviation.

MainProcess| 2024−05−20 07:29:50,490,490 multiprocessing INFO [User]
You will get a system description to code a differential equation simulator for.

System Description:```
Prediction of Treatment Response for Combined Chemo and Radiation Therapy for Non−Small Cell Lung Cancer Patients Using a Bio−
    Mathematical Model

Here you must model the state differential of tumor_volume, and chemotherapy_drug_concentration; with the input actions of .

Description of the variables:
∗ Volume of the tumor with units cm^3
∗ Concentration of the chemotherapy drug vinblastine with units mg/m^3

The time units is in days.

Additionally these variables have the ranges of:
∗ tumor_volume: [0.01433, 1170.861]
∗ chemotherapy_drug_concentration: [0, 9.9975]

The training dataset consists of 1000 patients, where each patient is observed for 60 days.
```

Modelling goals:```
∗ The parameters of the model will be optimized to an observed training dataset with the given simulator.
∗ The observed training dataset has very few samples, and the model must be able to generalize to unseen data.
```

Requirement Specification:```
∗ The code generated should achieve the lowest possible validation loss, of 1e−10 or less.
∗ The code generated should be interpretable, and fit the dataset as accurately as possible.
```

Skeleton code to fill in:```
class StateDifferential(nn.Module):
```

```
        def __init__(self):
            super(StateDifferential, self).__init__()
            # TODO: Fill in the code here

        def forward(self, tumor_volume: torch.Tensor, chemotherapy_drug_concentration: torch.Tensor) -> Tuple[torch.Tensor, torch.Tensor
            ]:
            # TODO: Fill in the code here
            return (d_tumor_volume__dt, d_chemotherapy_drug_concentration__dt)
```

Useful to know:```
* You are a code evolving machine, and you will be called 20 times to generate code, and improve the code to achieve the lowest
        possible validation loss.
* The model defines the state differential and will be used with an ODE solver to fit the observed training dataset.
* You can use any parameters you want and any black box neural network components (multi-layer perceptrons); however, you have to
        define these.
* It is preferable to decompose the system into differential equations (compartments) if possible.
* You can use any unary functions, for example log, exp, power etc.
* Under no circumstance can you change the skeleton code function definitions, only fill in the code.
* The input tensors are vectors of shape (batch_size).
* Use initially white box models first and then switch to hybrid white and black box models for the residuals, only after no further
        best program iteration improvement with white box models.
* Make sure your code follows the exact code skeleton specification.
* Use PyTorch.
```

Think step-by-step, and then give the complete full working code. You are generating code for iteration 0 out of 20.

MainProcess| 2024-05-20 07:29:50,490,490 multiprocessing INFO [User]
    You generated the following code completions, which then had their parameters optimized to the training dataset. Please reflect
        on how you can improve the code to minimize the validation loss to 1e-6 or less. The code examples are delineated by ###.

    Here are your previous iterations the best programs generated. Use it to see if you have exhausted white box models, i.e. when a
        white box model repeats with the same val loss and then only add black box models to the white box models:```
    Iteration 0. Best Val Loss: 43.21586608886719. Selected Features: tumor_volume, chemotherapy_drug_concentration. Model
        description: This is a white box model that uses a Gompertzian growth model for the tumor volume and a simple decay model
        for the chemotherapy drug concentration. The model includes parameters for the tumor growth rate and drug efficacy,
        which can be optimized based on the observed training dataset.
Iteration 1. Best Val Loss: 7.463240623474121. Selected Features: tumor_volume, chemotherapy_drug_concentration, radiotherapy_dosage
    . Model description: This is a white box model that uses a Gompertzian growth model for the tumor volume, a simple decay
    model for the chemotherapy drug concentration, and includes the effect of radiotherapy dosage. The model includes parameters
    for the tumor growth rate, drug efficacy, and radiotherapy efficacy, which can be optimized based on the observed training
    dataset.
    ```

    Please list in order of ranked importance (with the first feature to be acquired first) and the percentage of expected
        improvement of potential features to acquire next, such that the validation loss is minimized to 1e-6 or less. The
        possible additional features to select are: blood_type, smoking_status, patient_age, patient_weight, patient_height, sex,
        alcohol_consumption, previous_cancer_treatments, comorbidities, genetic_markers, chemotherapy_dosage,
        family_history_of_cancer, performance_status, dietary_habits, physical_activity_level, environmental_exposure,
        medication_adherence, psychological_stress_levels, socioeconomic_status, residential_location, support_network.

MainProcess| 2024-05-20 07:29:50,490,490 multiprocessing INFO [Progress: Step 1/0 | Retries: 0/30 | Token Capacity Used: 1.14% |
        Tokens remaining 126539]
MainProcess| 2024-05-20 07:30:23,219,219 multiprocessing INFO [LLM RESPONSE MESSAGE]
MainProcess| 2024-05-20 07:30:23,219,219 multiprocessing INFO [Function Call] [feature_acquisition] {
    "features": [
        {"name": "chemotherapy_dosage", "importance_score": 0.25},
        {"name": "genetic_markers", "importance_score": 0.20},
        {"name": "patient_age", "importance_score": 0.15},
        {"name": "previous_cancer_treatments", "importance_score": 0.10},
        {"name": "comorbidities", "importance_score": 0.10},
        {"name": "smoking_status", "importance_score": 0.05},
        {"name": "performance_status", "importance_score": 0.05},
        {"name": "medication_adherence", "importance_score": 0.05},
        {"name": "psychological_stress_levels", "importance_score": 0.05}
    ]
}
MainProcess| 2024-05-20 07:30:23,219,219 multiprocessing INFO [Selected new feature] SELECTED : chemotherapy_dosage Now using: ['
    tumor_volume', 'chemotherapy_drug_concentration', 'radiotherapy_dosage', 'chemotherapy_dosage']
MainProcess| 2024-05-20 07:30:23,222,222 multiprocessing INFO [System]
Objective: Write code to create an effective differential equation simulator for a given task.
Please note that the code should be fully functional. No placeholders.

You must act autonomously and you will receive no human input at any stage. You have to return as output the complete code for
        completing this task, and correctly improve the code to create the most accurate and realistic simulator possible.
You always write out the code contents. You always indent code with tabs.
You cannot visualize any graphical output. You exist within a machine. The code can include black box multi-layer perceptions where
        required.

Use the functions provided. When calling functions only provide a RFC8259 compliant JSON request following this format without
        deviation.

MainProcess| 2024-05-20 07:30:23,225,225 multiprocessing INFO [User]
You will get a system description to code a differential equation simulator for.

System Description:```
Prediction of Treatment Response for Combined Chemo and Radiation Therapy for Non-Small Cell Lung Cancer Patients Using a Bio-
        Mathematical Model

Here you must model the state differential of tumor_volume, and chemotherapy_drug_concentration; with the input actions of
        radiotherapy_dosage, chemotherapy_dosage.

Description of the variables:
* Volume of the tumor with units cm^3
* Concentration of the chemotherapy drug vinblastine with units mg/m^3
* Dosage of the radiotherapy with units Gy
* Dosage of the chemotherapy drug vinblastine with units mg/m^3

The time units is in days.

Additionally these variables have the ranges of:
* tumor_volume: [0.01433, 1170.861]
* chemotherapy_drug_concentration: [0, 9.9975]
* radiotherapy_dosage: [0, 2.0]
* chemotherapy_dosage: [0, 5.0]
```

The training dataset consists of 1000 patients, where each patient is observed for 60 days.
```

Modelling goals:```
* The parameters of the model will be optimized to an observed training dataset with the given simulator.
* The observed training dataset has very few samples, and the model must be able to generalize to unseen data.
```

Requirement Specification:```
* The code generated should achieve the lowest possible validation loss, of 1e−10 or less.
* The code generated should be interpretable, and fit the dataset as accurately as possible.
```

Skeleton code to fill in:```
```python
class StateDifferential(nn.Module):
    def __init__(self):
        super(StateDifferential, self).__init__()
        # TODO: Fill in the code here

    def forward(self, tumor_volume: torch.Tensor, chemotherapy_drug_concentration: torch.Tensor, radiotherapy_dosage: torch.Tensor,
            chemotherapy_dosage: torch.Tensor) -> Tuple[torch.Tensor, torch.Tensor]:
        # TODO: Fill in the code here
        return (d_tumor_volume__dt, d_chemotherapy_drug_concentration__dt)
```
```

Useful to know:```
* You are a code evolving machine, and you will be called 20 times to generate code, and improve the code to achieve the lowest possible validation loss.
* The model defines the state differential and will be used with an ODE solver to fit the observed training dataset.
* You can use any parameters you want and any black box neural network components (multi−layer perceptrons); however, you have to define these.
* It is preferable to decompose the system into differential equations (compartments) if possible.
* You can use any unary functions, for example log, exp, power etc.
* Under no circumstance can you change the skeleton code function definitions, only fill in the code.
* The input tensors are vectors of shape (batch_size).
* Use initially white box models first and then switch to hybrid white and black box models for the residuals, only after no further best program iteration improvement with white box models.
* Make sure your code follows the exact code skeleton specification.
* Use PyTorch.
```

Think step−by−step, and then give the complete full working code. You are generating code for iteration 2 out of 20.

MainProcess| 2024−05−20 07:30:23,225,225 multiprocessing INFO [User]
You generated the following code completions, which then had their parameters optimized to the training dataset. Please reflect on how you can improve the code to minimize the validation loss to 1e−6 or less. The code examples are delineated by ###.

Here are your previous iterations the best programs generated. Use it to see if you have exhausted white box models, i.e. when a white box model repeats with the same val loss and then only add black box models to the white box models:```
Iteration 0. Best Val Loss: 43.21586608886719. Selected Features: tumor_volume, chemotherapy_drug_concentration. Model description: This is a white box model that uses a Gompertzian growth model for the tumor volume and a simple decay model for the chemotherapy drug concentration. The model includes parameters for the tumor growth rate and drug efficacy, which can be optimized based on the observed training dataset.
Iteration 1. Best Val Loss: 7.463240623474121. Selected Features: tumor_volume, chemotherapy_drug_concentration, radiotherapy_dosage. Model description: This is a white box model that uses a Gompertzian growth model for the tumor volume, a simple decay model for the chemotherapy drug concentration, and includes the effect of radiotherapy dosage. The model includes parameters for the tumor growth rate, drug efficacy, and radiotherapy efficacy, which can be optimized based on the observed training dataset.
```

Here are the top code completions so far that you have generated, sorted for the lowest validation loss last:```

Val Loss: 43.2 (Where the val loss per dimension is tumor_volume val loss: 79, chemotherapy_drug_concentration val loss: 7.42)
        Iteration: 0
###
```python
import torch
import torch.nn as nn
from typing import Tuple

class StateDifferential(nn.Module):
    def __init__(self):
        super(StateDifferential, self).__init__()
        # Define the parameters for the tumor growth model
        self.alpha = nn.Parameter(torch.tensor(0.1))  # Tumor growth rate
        self.beta = nn.Parameter(torch.tensor(0.01)) # Drug efficacy

    def forward(self, tumor_volume: torch.Tensor, chemotherapy_drug_concentration: torch.Tensor) -> Tuple[torch.Tensor, torch.Tensor]:
        # Tumor growth model based on Gompertzian growth
        d_tumor_volume__dt = self.alpha * torch.log(1170.861 / tumor_volume) * tumor_volume
        # Drug effect on tumor volume
        d_tumor_volume__dt -= self.beta * chemotherapy_drug_concentration * tumor_volume

        # Assuming constant drug concentration decay
        drug_decay_rate = 0.1
        d_chemotherapy_drug_concentration__dt = -drug_decay_rate * chemotherapy_drug_concentration

        return (d_tumor_volume__dt, d_chemotherapy_drug_concentration__dt)
```
optimized_parameters = {'alpha': 0.00944874994456768, 'beta': 0.031064828857779503}
###

Val Loss: 7.46 (Where the val loss per dimension is tumor_volume val loss: 7.51, chemotherapy_drug_concentration val loss: 7.42)
        Iteration: 1
###
```python
import torch
import torch.nn as nn
from typing import Tuple

class StateDifferential(nn.Module):
    def __init__(self):
        super(StateDifferential, self).__init__()
```

```
        # Define the parameters for the tumor growth model
        self.alpha = nn.Parameter(torch.tensor(0.00944874994456768))  # Optimized tumor growth rate
        self.beta = nn.Parameter(torch.tensor(0.031064828857779503)) # Optimized drug efficacy
        # Define the parameter for the radiotherapy effect
        self.gamma = nn.Parameter(torch.tensor(0.1)) # Radiotherapy efficacy

    def forward(self, tumor_volume: torch.Tensor, chemotherapy_drug_concentration: torch.Tensor, radiotherapy_dosage: torch.Tensor)
            -> Tuple[torch.Tensor, torch.Tensor]:
        # Tumor growth model based on Gompertzian growth
        d_tumor_volume__dt = self.alpha * torch.log(1170.861 / tumor_volume) * tumor_volume
        # Drug effect on tumor volume
        d_tumor_volume__dt -= self.beta * chemotherapy_drug_concentration * tumor_volume
        # Radiotherapy effect on tumor volume
        d_tumor_volume__dt -= self.gamma * radiotherapy_dosage * tumor_volume

        # Assuming constant drug concentration decay
        drug_decay_rate = 0.1
        d_chemotherapy_drug_concentration__dt = -drug_decay_rate * chemotherapy_drug_concentration

        return (d_tumor_volume__dt, d_chemotherapy_drug_concentration__dt)
```
optimized_parameters = {'alpha': 0.025857042521238327, 'beta': 0.025862518697977066, 'gamma': 0.03730243816971779}
###

```

Code Skeleton:```
class StateDifferential(nn.Module):
    def __init__(self):
        super(StateDifferential, self).__init__()
        # TODO: Fill in the code here

    def forward(self, tumor_volume: torch.Tensor, chemotherapy_drug_concentration: torch.Tensor, radiotherapy_dosage: torch.Tensor,
            chemotherapy_dosage: torch.Tensor) -> Tuple[torch.Tensor, torch.Tensor]:
        # TODO: Fill in the code here
        return (d_tumor_volume__dt, d_chemotherapy_drug_concentration__dt)
```

Please reflect on how you can improve the code to fit the dataset as accurately as possible, and be interpretable. Use the code
    skeleton above to provide the code. Please also specify only the input parameters that you want to include in the model in
    from the skeleton code. Think step-by-step. Where applicable use the values of the optimized parameters to reason how the
    code can be improved to fit the dataset as accurately as possible. This is for generating new code for the next iteration 2
    out of 20.

MainProcess| 2024-05-20 07:30:23,225,225 multiprocessing INFO [Progress: Step 1/0 | Retries: 0/30 | Token Capacity Used: 1.87% |
    Tokens remaining 125608]
MainProcess| 2024-05-20 07:33:55,671,671 multiprocessing INFO [LLM RESPONSE MESSAGE]
MainProcess| 2024-05-20 07:33:55,671,671 multiprocessing INFO [Function Call] [complete_StateDifferential_code] {
    "StateDifferential_code": "import torch\nimport torch.nn as nn\nfrom typing import Tuple\n\nclass StateDifferential(nn.Module):\
        n    def __init__(self):\n        super(StateDifferential, self).__init__()\n        # Define the parameters for the
        tumor growth model\n        self.alpha = nn.Parameter(torch.tensor(0.025857042521238327))  # Optimized tumor growth rate\
        n        self.beta = nn.Parameter(torch.tensor(0.025862518697977066)) # Optimized drug efficacy\n        # Define the
        parameter for the radiotherapy effect\n        self.gamma = nn.Parameter(torch.tensor(0.03730243816971779)) # Optimized
        radiotherapy efficacy\n        # Define the parameter for the chemotherapy effect\n        self.delta = nn.Parameter(
        torch.tensor(0.1)) # Chemotherapy dosage efficacy\n\n    def forward(self, tumor_volume: torch.Tensor,
        chemotherapy_drug_concentration: torch.Tensor, radiotherapy_dosage: torch.Tensor, chemotherapy_dosage: torch.Tensor) ->
        Tuple[torch.Tensor, torch.Tensor]:\n        # Tumor growth model based on Gompertzian growth\n        d_tumor_volume__dt
        = self.alpha * torch.log(1170.861 / tumor_volume) * tumor_volume\n        # Drug effect on tumor volume\n
        d_tumor_volume__dt -= self.beta * chemotherapy_drug_concentration * tumor_volume\n        # Radiotherapy effect on tumor
        volume\n        d_tumor_volume__dt -= self.gamma * radiotherapy_dosage * tumor_volume\n        # Chemotherapy dosage
        effect on tumor volume\n        d_tumor_volume__dt -= self.delta * chemotherapy_dosage * tumor_volume\n\n        #
        Assuming constant drug concentration decay\n        drug_decay_rate = 0.1\n        d_chemotherapy_drug_concentration__dt
        = -drug_decay_rate * chemotherapy_drug_concentration\n\n        return (d_tumor_volume__dt,
        d_chemotherapy_drug_concentration__dt)",
    "code_description": "This is a white box model that uses a Gompertzian growth model for the tumor volume, a simple decay model
        for the chemotherapy drug concentration, and includes the effects of both radiotherapy and chemotherapy dosages. The
        model includes parameters for the tumor growth rate, drug efficacy, radiotherapy efficacy, and chemotherapy dosage
        efficacy, which can be optimized based on the observed training dataset.",
    "model_input_variables": ["tumor_volume", "chemotherapy_drug_concentration", "radiotherapy_dosage", "chemotherapy_dosage"]
}
MainProcess| 2024-05-20 07:34:06,547,547 multiprocessing INFO Early stopping triggered at epoch 1120
MainProcess| 2024-05-20 07:34:06,560,560 multiprocessing INFO [D3-feature-select | Cancer-random-features | 12][Generation 2] | Top
    Fitness: 6.335662364959717 | Num Programs: 3 | Mean Fitness: 19.00492302576701 | Fitnesses: [6.335662364959717,
    7.463240623474121, 43.21586608886719] | Current Gen Val Loss: 6.335662364959717
MainProcess| 2024-05-20 07:34:06,560,560 multiprocessing INFO [D3-feature-select | Cancer-random-features | 12][GEN RESULT] {'method
    ': 'D3-feature-select', 'env_name': 'Cancer-random-features', 'seed': 12, 'generation': 2, 'top_fitness': 6.335662364959717,
    'num_programs': 3, 'mean_fitness': 19.00492302576701, 'fitnesses': [6.335662364959717, 7.463240623474121, 43.21586608886719],
    'current_gen_val_loss': 6.335662364959717}
MainProcess| 2024-05-20 07:34:06,561,561 multiprocessing INFO [System]

Objective: Write code to create an effective differential equation simulator for a given task.
Please note that the code should be fully functional. No placeholders.

You must act autonomously and you will receive no human input at any stage. You have to return as output the complete code for
    completing this task, and correctly improve the code to create the most accurate and realistic simulator possible.
You always write out the code contents. You always indent code with tabs.
You cannot visualize any graphical output. You exist within a machine. The code can include black box multi-layer perceptions where
    required.

Use the functions provided. When calling functions only provide a RFC8259 compliant JSON request following this format without
    deviation.

MainProcess| 2024-05-20 07:34:06,561,561 multiprocessing INFO [User]

You will get a system description to code a differential equation simulator for.

System Description:```
Prediction of Treatment Response for Combined Chemo and Radiation Therapy for Non-Small Cell Lung Cancer Patients Using a Bio-
    Mathematical Model

Here you must model the state differential of tumor_volume, and chemotherapy_drug_concentration; with the input actions of .

Description of the variables:
* Volume of the tumor with units cm^3
* Concentration of the chemotherapy drug vinblastine with units mg/m^3

The time units is in days.

Additionally these variables have the ranges of:
* tumor_volume: [0.01433, 1170.861]
* chemotherapy_drug_concentration: [0, 9.9975]

The training dataset consists of 1000 patients, where each patient is observed for 60 days.
```

Modelling goals:```
* The parameters of the model will be optimized to an observed training dataset with the given simulator.
* The observed training dataset has very few samples, and the model must be able to generalize to unseen data.
```

Requirement Specification:```
* The code generated should achieve the lowest possible validation loss, of 1e−10 or less.
* The code generated should be interpretable, and fit the dataset as accurately as possible.
```

Skeleton code to fill in:```
```python
class StateDifferential(nn.Module):
    def __init__(self):
        super(StateDifferential, self).__init__()
        # TODO: Fill in the code here

    def forward(self, tumor_volume: torch.Tensor, chemotherapy_drug_concentration: torch.Tensor) -> Tuple[torch.Tensor, torch.Tensor]:
        # TODO: Fill in the code here
        return (d_tumor_volume__dt, d_chemotherapy_drug_concentration__dt)
```
```

Useful to know:```
* You are a code evolving machine, and you will be called 20 times to generate code, and improve the code to achieve the lowest possible validation loss.
* The model defines the state differential and will be used with an ODE solver to fit the observed training dataset.
* You can use any parameters you want and any black box neural network components (multi−layer perceptrons); however, you have to define these.
* It is preferable to decompose the system into differential equations (compartments) if possible.
* You can use any unary functions, for example log, exp, power etc.
* Under no circumstance can you change the skeleton code function definitions, only fill in the code.
* The input tensors are vectors of shape (batch_size).
* Use initially white box models first and then switch to hybrid white and black box models for the residuals, only after no further best program iteration improvement with white box models.
* Make sure your code follows the exact code skeleton specification.
* Use PyTorch.
```

Think step−by−step, and then give the complete full working code. You are generating code for iteration 0 out of 20.

MainProcess| 2024−05−20 07:34:06,561,561 multiprocessing INFO [User]
    You generated the following code completions, which then had their parameters optimized to the training dataset. Please reflect on how you can improve the code to minimize the validation loss to 1e−6 or less. The code examples are delineated by ###.

    Here are your previous iterations the best programs generated. Use it to see if you have exhausted white box models, i.e. when a white box model repeats with the same val loss and then only add black box models to the white box models:```
    Iteration 0. Best Val Loss: 43.21586608886719. Selected Features: tumor_volume, chemotherapy_drug_concentration. Model description: This is a white box model that uses a Gompertzian growth model for the tumor volume and a simple decay model for the chemotherapy drug concentration. The model includes parameters for the tumor growth rate and drug efficacy, which can be optimized based on the observed training dataset.
    Iteration 1. Best Val Loss: 7.463240623474121. Selected Features: tumor_volume, chemotherapy_drug_concentration, radiotherapy_dosage. Model description: This is a white box model that uses a Gompertzian growth model for the tumor volume, a simple decay model for the chemotherapy drug concentration, and includes the effect of radiotherapy dosage. The model includes parameters for the tumor growth rate, drug efficacy, and radiotherapy efficacy, which can be optimized based on the observed training dataset.
    Iteration 2. Best Val Loss: 6.335662364959717. Selected Features: tumor_volume, chemotherapy_drug_concentration, radiotherapy_dosage, chemotherapy_dosage. Model description: This is a white box model that uses a Gompertzian growth model for the tumor volume, a simple decay model for the chemotherapy drug concentration, and includes the effects of both radiotherapy and chemotherapy dosages. The model includes parameters for the tumor growth rate, drug efficacy, radiotherapy efficacy, and chemotherapy dosage efficacy, which can be optimized based on the observed training dataset.
    ```

    Please list in order of ranked importance (with the first feature to be acquired first) and the percentage of expected improvement of potential features to acquire next, such that the validation loss is minimized to 1e−6 or less. The possible additional features to select are: blood_type, smoking_status, patient_age, patient_weight, patient_height, sex, alcohol_consumption, previous_cancer_treatments, comorbidities, genetic_markers, family_history_of_cancer, performance_status, dietary_habits, physical_activity_level, environmental_exposure, medication_adherence, psychological_stress_levels, socioeconomic_status, residential_location, support_network.

MainProcess| 2024−05−20 07:34:06,561,561 multiprocessing INFO [Progress: Step 1/0 | Retries: 0/30 | Token Capacity Used: 1.23% | Tokens remaining 126425]
MainProcess| 2024−05−20 07:34:55,703,703 multiprocessing INFO [LLM RESPONSE MESSAGE]
MainProcess| 2024−05−20 07:34:55,704,704 multiprocessing INFO [Function Call] [feature_acquisition] {
```json
"features": [
    {
        "name": "genetic_markers",
        "importance_score": 0.25
    },
    {
        "name": "previous_cancer_treatments",
        "importance_score": 0.2
    },
    {
        "name": "comorbidities",
        "importance_score": 0.15
    },
    {
        "name": "patient_age",
        "importance_score": 0.1
    },
    {
        "name": "smoking_status",
        "importance_score": 0.1
    },
    {
        "name": "performance_status",
        "importance_score": 0.05
    },
    {
```

```
            "name": "medication_adherence",
            "importance_score": 0.05
        },
        {
            "name": "psychological_stress_levels",
            "importance_score": 0.05
        },
        {
            "name": "environmental_exposure",
            "importance_score": 0.05
        }
    ]
}
```

MainProcess| 2024−05−20 07:34:55,704,704 multiprocessing INFO [Selected new feature] SELECTED : genetic_markers Now using: ['tumor_volume', 'chemotherapy_drug_concentration', 'radiotherapy_dosage', 'chemotherapy_dosage', 'genetic_markers']

## G.3    Other Datasets Discovered White-Box Models

In the main paper we presented the best Warfarin discovered white-box model, however after running D3-white-box across all the other datasets, we include a few of the white-box models that it discovered and generated for each other respective dataset used that we compared against. Of particular interest, is that these white-box models fit the dataset well, and can even outperform some of the other neural network based baselines—we highlight that with the right functional form of the white-box model, we can achieve better performance than a largely over-parameterized neural network model, that has signficiantly more parameters (such as an RNN with 18,000 parameters), than a white-box model with 5-15 parameters.

### Lung Cancer

```
Val Loss: 0.352 (Where the val loss per dimension is tumor_volume val loss: 0.352) Iteration: 7
###
```
class StateDifferential(nn.Module):
    def __init__(self):
        super(StateDifferential, self).__init__()
        # Define the parameters for the tumor growth model
        # Initialize parameters with values closer to the optimized values from iteration 6
        self.alpha = nn.Parameter(torch.tensor(0.0028))
        self.beta = nn.Parameter(torch.tensor(1000.33))
        # Initialize gamma as a small positive value
        self.gamma = nn.Parameter(torch.tensor(0.001))
        # Initialize delta with a value closer to the optimized value from iteration 6
        self.delta = nn.Parameter(torch.tensor(0.705))
        # Introduce epsilon to represent a new biological term, e.g., immune response
        self.epsilon = nn.Parameter(torch.tensor(0.001))

    def forward(self, tumor_volume: torch.Tensor) -> Tuple[torch.Tensor]:
        # Ensure tumor_volume is non−negative
        tumor_volume = torch.clamp(tumor_volume, min=0)
        # Calculate the growth term, necrosis term, angiogenesis term, and new biological term
        growth_term = self.alpha * tumor_volume * (1 − tumor_volume / self.beta)
        necrosis_term = −self.gamma * tumor_volume
        angiogenesis_term = self.delta * torch.sqrt(tumor_volume)
        immune_response_term = −self.epsilon * tumor_volume
        # Rate of change of tumor volume
        d_tumor_volume__dt = growth_term + necrosis_term + angiogenesis_term + immune_response_term
        return (d_tumor_volume__dt,)
```
optimized_parameters = {'alpha': 0.0026573685463517904, 'beta': 1000.9459838867188, 'gamma': −0.007219356019049883, 'delta': 0.7442551851272583, 'epsilon': −0.007219356019049883}
###
```

### Lung Cancer (with Chemo.)

```
Val Loss: 2.45 (Where the val loss per dimension is tumor_volume val loss: 4.91, chemotherapy_drug_concentration val loss: 2.68e−06)
    Iteration: 0
###
```
class StateDifferential(nn.Module):
    def __init__(self):
        super(StateDifferential, self).__init__()
        # Define the parameters for the model
        self.k_growth = nn.Parameter(torch.tensor(0.1))
        self.k_decay = nn.Parameter(torch.tensor(0.1))
        self.k_chemo_effect = nn.Parameter(torch.tensor(0.1))

    def forward(self, tumor_volume: torch.Tensor, chemotherapy_drug_concentration: torch.Tensor, chemotherapy_dosage: torch.Tensor)
        -> Tuple[torch.Tensor, torch.Tensor]:
        # Calculate the rate of change of tumor volume
        d_tumor_volume__dt = self.k_growth * tumor_volume − self.k_chemo_effect * chemotherapy_drug_concentration * tumor_volume

        # Calculate the rate of change of chemotherapy drug concentration
        d_chemotherapy_drug_concentration__dt = chemotherapy_dosage − self.k_decay * chemotherapy_drug_concentration

        return (d_tumor_volume__dt, d_chemotherapy_drug_concentration__dt)
```
optimized_parameters = {'k_growth': 0.04456980526447296, 'k_decay': 0.4996906816959381, 'k_chemo_effect': 0.027509577572345734}
###
```

### Lung Cancer (with Chemo. & Radio.)

```
Val Loss: 1.42 (Where the val loss per dimension is tumor_volume val loss: 2.85, chemotherapy_drug_concentration val loss: 1.96e−11)
    Iteration: 0
```

###
```
import torch
import torch.nn as nn
from typing import Tuple

class StateDifferential(nn.Module):
    def __init__(self):
        super(StateDifferential, self).__init__()
        # Define the parameters for the model
        self.k_growth = nn.Parameter(torch.tensor(0.1))
        self.k_decay = nn.Parameter(torch.tensor(0.1))
        self.k_chemo_effect = nn.Parameter(torch.tensor(-0.05))
        self.k_radio_sensitivity = nn.Parameter(torch.tensor(-0.02))

    def forward(self, tumor_volume: torch.Tensor, chemotherapy_drug_concentration: torch.Tensor, chemotherapy_dosage: torch.Tensor,
            radiotherapy_dosage: torch.Tensor) -> Tuple[torch.Tensor, torch.Tensor]:
        # Calculate the differential of tumor volume
        d_tumor_volume__dt = self.k_growth * tumor_volume - self.k_chemo_effect * chemotherapy_drug_concentration * tumor_volume -
                self.k_radio_sensitivity * radiotherapy_dosage * tumor_volume

        # Calculate the differential of chemotherapy drug concentration
        d_chemotherapy_drug_concentration__dt = chemotherapy_dosage - self.k_decay * chemotherapy_drug_concentration

        return (d_tumor_volume__dt, d_chemotherapy_drug_concentration__dt)
```
optimized_parameters = {'k_growth': 0.04546591639518738, 'k_decay': 0.49999910593032837, 'k_chemo_effect': 0.027269458398222923, '
        k_radio_sensitivity': 0.04899420961737633}
###

## Plankton Microcosm

Val Loss: 5.76e−06 (Where the val loss per dimension is prey_population val loss: 1.18e−05, intermediate_population val loss: 1.37e
        −06, top_predators_population val loss: 4.07e−06) Iteration: 13
###
```
class StateDifferential(nn.Module):
    def __init__(self):
        super(StateDifferential, self).__init__()
        # Define the parameters for the model with initial values close to optimized parameters
        self.alpha = nn.Parameter(torch.tensor(0.04))   # Prey growth rate
        self.beta = nn.Parameter(torch.tensor(0.23))    # Prey death rate due to predation
        self.gamma = nn.Parameter(torch.tensor(0.23))   # Intermediate predator growth rate
        self.delta = nn.Parameter(torch.tensor(0.13))   # Intermediate predator death rate
        self.eta = nn.Parameter(torch.tensor(0.06))     # Top predator growth rate
        self.theta = nn.Parameter(torch.tensor(-0.12))  # Top predator death rate
        self.kappa = nn.Parameter(torch.tensor(2.2))    # Prey carrying capacity
        self.sigma = nn.Parameter(torch.tensor(1.4))    # Intermediate predator carrying capacity
        self.rho = nn.Parameter(torch.tensor(0.7))      # Top predator carrying capacity
        self.omega = nn.Parameter(torch.tensor(0.02))   # Effect of top predators on intermediate predator mortality
        self.psi = nn.Parameter(torch.tensor(0.12))     # Effect of prey abundance on intermediate predator growth
        self.phi = nn.Parameter(torch.tensor(0.03))     # Nonlinear interaction term for prey and intermediate predators
        self.xi = nn.Parameter(torch.tensor(0.03))      # Nonlinear interaction term for intermediate predators and top predators

    def forward(self, prey_population: torch.Tensor, intermediate_population: torch.Tensor, top_predators_population: torch.Tensor)
            -> Tuple[torch.Tensor, torch.Tensor, torch.Tensor]:
        # Prey growth limited by carrying capacity
        d_prey_population__dt = self.alpha * prey_population * (1 - prey_population / self.kappa) - self.beta * prey_population *
                intermediate_population
        # Intermediate predator growth with nonlinear interaction term
        d_intermediate_population__dt = self.gamma * intermediate_population * (1 - intermediate_population / self.sigma) * (self.
                psi * prey_population) - self.delta * intermediate_population * top_predators_population - self.phi * prey_population
                **2 * intermediate_population
        # Top predator growth with effect of intermediate predators and nonlinear interaction term
        d_top_predators_population__dt = self.eta * top_predators_population * (1 - top_predators_population / self.rho) - self.
                theta * top_predators_population * intermediate_population - self.xi * intermediate_population**2 *
                top_predators_population

        return (d_prey_population__dt, d_intermediate_population__dt, d_top_predators_population__dt)
```
optimized_parameters = {'alpha': 0.04999999329447746, 'beta': 0.23999999463558197, 'gamma': 0.239999920129776, 'delta':
        0.1399998569488525, 'eta': 0.05000030994415283, 'theta': -0.1299992799758911, 'kappa': 2.2099997997283936, 'sigma':
        1.3900007009506226, 'rho': 0.6900002956390381, 'omega': 0.019999999552965164, 'psi': 0.12999995052814484, 'phi':
        0.020000003278255463, 'xi': 0.020000090822577477}
###

# H   Future Work

The Data-Driven Discovery (D3) framework demonstrates initial success in generating interpretable pharmacokinetic (PK) models using an LLM-based iterative process. However, future work can expand and deepen the D3 approach in multiple directions:

1. **Tree-Based and Graph-Based Generation Strategies:** Extending D3 with tree-based generation strategies, such as Tree of Thought [Yao et al., 2024] or Graph of Thought [Besta et al., 2024], could provide a more structured exploration of model space. Such approaches could enhance D3's capacity to generate increasingly complex models over time, aligning with a growing need for adaptive modeling frameworks that account for intricate biological dynamics and drug interactions.

2. **Robustness Through Retrieval-Augmented Generation:** Incorporating Retrieval-Augmented Generation (RAG) techniques could bolster D3's robustness by enabling on-demand access to external domain knowledge. RAG techniques [Lewis et al., 2020] can

serve as a mechanism to manage LLM hallucinations and ensure that model suggestions reflect state-of-the-art pharmacological principles. This capability would minimize reliance on the initial system descriptions and provide more flexible, context-driven model adjustments.

3. **Adaptive Feature Acquisition:** The current D3 implementation assumes that newly acquired features apply uniformly to all individuals in a dataset, which may not be feasible in practice due to data variability and differing acquisition costs across individuals. Future iterations could focus on more nuanced feature acquisition strategies that adapt to individual data constraints, potentially exploring patient-specific feature prioritization algorithms to reduce acquisition costs in clinical settings.

4. **Enhanced Tool Use by LLM:** While D3 utilizes GPT-4 for model discovery, integrating more advanced tool usage capabilities could further enhance its performance. Developing mechanisms for LLMs to independently call and evaluate external computational tools could extend D3's evaluation capacity, leading to richer and more rigorous model feedback, particularly for complex, multi-stage PK processes.

5. **Extending Applications to Multi-Phase PK/PD Studies:** Expanding D3's applicability to multi-phase PK/PD studies, such as those involving dynamic drug interaction effects or multi-compartmental absorption models, could broaden the impact of the framework. These applications would push D3 toward modeling scenarios with temporal and interventional complexity, where capturing delayed and interactive effects becomes paramount.

These advancements would not only strengthen D3's theoretical underpinnings but also enhance its practical applicability in clinical pharmacology, potentially contributing to improved drug efficacy and safety.

# I  Additional Sections

## I.1  Cost Function Considerations

In our feature acquisition process, the cost function $l(h_i)$ plays a central role in determining the feasibility of acquiring new features. This function captures the computational, logistical, and practical costs associated with acquiring additional data for model refinement, thereby guiding the D3 framework to make efficient, data-driven decisions.

While our primary focus has been on computational costs and feature utility, it is important to recognize that feature acquisition often involves tradeoffs, particularly when biomarkers or other data types are costly, time-consuming, or invasive to obtain. We outline several key considerations and potential tradeoffs below:

1. **Cost-Benefit Analysis of Biomarker Acquisition**: In pharmacokinetic (PK) modeling, biomarkers can provide high-impact data that significantly improve model accuracy. However, some biomarkers (e.g., genetic markers, certain plasma proteins) are costly or require invasive procedures. Thus, the acquisition cost $l(h_i)$ should reflect not only monetary or computational expenses but also potential patient discomfort and clinical feasibility. This factor can be critical when evaluating the value of acquiring such features in real-world applications.

2. **Marginal Utility and Redundancy Avoidance**: Including a cost function allows D3 to weigh the marginal utility of new features against their acquisition cost, thereby avoiding redundancy. For instance, if a biomarker's expected contribution to model improvement is low, D3 can prioritize less costly or more readily available features, optimizing overall efficiency. This approach minimizes unnecessary data collection, thus reducing the risk of overfitting and keeping model complexity manageable.

3. **Scalability of Feature Acquisition**: In large-scale clinical studies, acquiring high-cost features across a population sample may be prohibitive. D3's cost function can be adjusted to prioritize scalable data collection strategies by favoring features with lower per-patient costs or those that require fewer resources to standardize across study participants. This balance enables broader, real-world applicability while still allowing D3 to leverage high-value data where feasible.

4. **Dynamic Cost Function Adaptation**: As D3 iterates and learns from acquired features, the cost function $l(h_i)$ can dynamically adapt based on the observed impact of previous acquisitions. For example, if certain features consistently yield low model improvement relative to cost, the function can adjust to deprioritize similar features. Such dynamic adaptation helps D3 achieve a more nuanced balance between cost and utility as it iteratively refines the model.

These cost function considerations are integral to D3's feature acquisition approach, supporting a more balanced model refinement process that aligns predictive accuracy with practical feasibility. Future iterations of D3 could further incorporate adaptive mechanisms to fine-tune these tradeoffs, potentially optimizing feature acquisition across varying clinical and resource settings.

