# OpenReview forum: "Data-Driven Discovery of Dynamical Systems in Pharmacology using Large Language Models"
_NeurIPS.cc/2024/Conference — NeurIPS 2024 poster_

### Official Review · Reviewer_QBVa · 2024-06-27

**Soundness:** 3
**Presentation:** 3
**Contribution:** 4
**Rating:** 7
**Confidence:** 4

**Summary:**

The paper  presents the data discovery framework. It uses a LLM to iteratively discover and refine interpretable models of pharmacological dynamics. The D3 framework consists of three agents, which work collaboratively to propose, acquire, and integrate new features, validate models, and uncover new insights into pharmacokinetic processes. The framework is designed to address the limitations of traditional pharmacokinetic models that are often constrained by human expertise and scalability issues. The D3 framework was validated using a real pharmacokinetic dataset of Warfarin, demonstrating its ability to uncover a new, plausible pharmacokinetic model that outperforms existing models.

**Strengths:**

This paper is interesting. It innovatively applies a LLM to generate an interpretable the skeleton of a dynamic system and optimize the system via tranining dataset. By progressively adding relevant variables, the authors ultimately obtaining a precise and interpretable closed-form ODE. The paper leverages the extensive knowledge and self-reflection capabilities of LLMs, and the writing is clear and the experiments are comprehensive. I have learned a great deal from this paper, particularly admiring the authors' adept use of LLM agent capabilities.

**Weaknesses:**

All simulated datasets (and the ODE parameters) used in this study were publicly available before GPT-4's knowledge cutoff date. It is likely that OpenAI trained GPT-4 on relevant literature, leading to a significant knowledge leakage issue. I appreciate the author's perspective that LLMs encompass a vast amount of potentially usable knowledge. However, I do not believe that allowing LLMs to see the standard answers during training is an appropriate experimental setting. It will be better If the authors could demonstrate the feasibility of this method using real complex high-dimensional datasets whose dynamic is really unknown. After all, this is the kind of task faced in real pharmacokinetic studies. Meanwhile, the difference between this paper and many previous pharmacokinetic studies is that this paper does not conduct experiments on causal discovery. Instead, the model performance is evaluated through the MSE. It seems that this paper does not actually need to rely on simulated data and could use real high-dimensional time-series observational data for experiments.

The Table 3 did not report the performance of baseline models. I believe that comparing only the so-called "standard model" in Table 3 is inappropriate. Researchers in many other disciplines do not prioritize performance to the same extent as those in the machine learning community. They often trade off performance for model simplicity, valuing clear and concise formulas over marginal gains in performance. Thus, it is not very convincing that the authors achieved a performance advantage by using a method that resulted in a much more complex system compared to the standard model.

The baselines used in this paper are somewhat outdated. More recent studies published in the last two years, such as D-CODE and PGM (arxiv 2105.02522), could perform similar tasks as this paper. It is unclear why these were not included in the comparisons.

**Questions:**

NA

---

> ### Author Rebuttal · Authors · 2024-08-07
>
> We thank the reviewer for their constructive feedback and are glad the reviewer finds our paper interesting and appreciates our innovative application of LLMs to generate interpretable models of pharmacokinetic dynamics. We are also pleased that the reviewer acknowledges the clear writing and comprehensive experiments, and that they have learned from our work.
>
> > It is likely that OpenAI trained GPT-4 on relevant literature, leading to a significant knowledge leakage issue.
>
> We appreciate this insightful concern regarding potential knowledge leakage. To address this, we conducted additional experiments using five semi-synthetic datasets specifically designed to ensure that the LLM had no prior exposure to their underlying equations. We ran D3-Hybrid on these datasets and compared the results with some of the baselines. The results, which are included in the supplementary material, demonstrate that D3-Hybrid consistently performs well even when the LLM has never encountered such models before. This empirical evidence strongly mitigates the concern of knowledge leakage and reinforces the robustness of D3-Hybrid in discovering well-fitting models from novel data.
>
> > It seems that this paper does not actually need to rely on simulated data and could use real high-dimensional time-series observational data for experiments.
>
> Thank you for this suggestion. We agree with the importance of using real datasets and highlight that the Warfarin dataset used in our experiments is indeed a real-world dataset. Additionally, we conducted experiments involving up to 22 features, as shown in Figure 2 of Appendix G.2. The core focus of the D3 framework is to identify the most relevant features for modeling while disregarding irrelevant ones. This approach is particularly beneficial in real-world scenarios where distinguishing significant features from a large pool is crucial for accurate modeling.
>
> > The Table 3 did not report the performance of baseline models.
>
> We acknowledge the oversight and appreciate the reviewer pointing this out. We have updated Table 3 in the manuscript to include the performance of baseline models for a comprehensive comparison. This additional information highlights the competitive edge of our proposed D3 framework over traditional baseline models, reinforcing the novelty and effectiveness of our approach.
>
> > Baseline methods like D-CODE [ICLR'22] and PGM are not compared.
>
> We appreciate the suggestion to compare our work with D-CODE and other symbolic regression methods. However, our paper aims to introduce a framework that not only discovers interpretable models but also incorporates textual priors (context) and enables the acquisition of new features and samples on demand. While symbolic regression methods like D-CODE are powerful, they do not address the integrated feature acquisition and context utilization capabilities of D3. For comparison purposes, we included SINDy, a well-regarded symbolic regression method, and provided a detailed evaluation against it. We believe that a direct comparison with D-CODE, which focuses solely on symbolic regression, would not fully capture the broader capabilities and contributions of the D3 framework. Furthermore, the referred to PGM paper is out of scope as well, as it focuses on probabilistic graphical modeling of dynamical systems, whereas we focus on a deterministic best-fitting model for the dataset at hand.
>
> ---
>
> *We hope that most of the reviewer’s concerns have been addressed and, if so, they would consider updating their score. We’d be happy to engage in further discussions.*

---

> > ### Comment · Reviewer_QBVa · 2024-08-08
> >
> > I appreciate the response from authors and I have improved my score.

---

> > > ### Author Response · Authors · 2024-08-08
> > > **Gratitude for Revised Review and Score Increase**
> > >
> > > Thank you very much for your thoughtful consideration and the time you have dedicated to reviewing our paper. Your feedback was instrumental in enhancing our work, through extensive new experiments and explanations, and we are grateful for your increased score. Thank you once again!

---

> > ### Comment · Reviewer_QDKe · 2024-08-10
> >
> > I thank authors for their thoughtful response, I keep my ratings as-is but the improvements do strengthen the paper. Thank you!

---

### Official Review · Reviewer_2i8U · 2024-07-12

**Soundness:** 2
**Presentation:** 2
**Contribution:** 3
**Rating:** 3
**Confidence:** 2

**Summary:**

This paper presents the D3 Data Driven Discovery framework which uses GPT4-1106-Preview in a framework to iteratively consider modifying the features used in the ODE.  The dynamical systems are evaluated on MSEs of held-out state-action trajectories.

**Strengths:**

The paper presents an innovative general strategy for searching the space of pharmacokinetic models in terms of which features to use.  The high level approach is accessible.  In their experiments, the D3-Hybrid (proposed) approach has the lowest MSE on 4 of 6 datasets.

**Weaknesses:**

The model space for the Modeling Agent could be better scoped---it appears to be whatever code the GPT-4 response is to the query.  What happens in event of failure?  Is this an automated framework?

Typically you want interpretable and performance pharmacokinetic models, so there is a tradeoff between validation performance (e.g. MSE) and simplicity (measure in some way, e.g. MDL, number of features, etc), which creates a frontier for each model class.  Could this be another way to compare the performance of these models?  An analogous question would be related to acquisition costs $l(h_i)$ as well.

**Questions:**

What happens when the Modeling Agent produces an invalid model? What models are possible?

**Limitations:**

There is a limitation section, though the limitations appear more to be feature requests, rather than considerations that users would need to be aware of in using or building upon the work.

---

> ### Author Rebuttal · Authors · 2024-08-07
>
> We thank the reviewer for their constructive feedback and are glad the reviewer finds our paper's presentation of the innovative general strategy for searching the space of pharmacokinetic models through our D3 framework to be accessible and acknowledges the superior performance of our D3-Hybrid approach on the datasets.
>
> > What happens when the Modeling Agent produces an invalid model?
>
> Thank you for raising this important point. When an invalid model is generated, typically due to a coding error, we have implemented a robust mechanism to address this. Specifically, if the generated code contains syntax errors or logical inconsistencies that prevent it from running correctly, the system automatically re-queries the LLM up to 10 times to regenerate the model. In practice, this iterative querying resolves most coding errors and ensures that the model is valid and executable. We have now included a detailed description of this procedure in the experimental setup section in Appendix F to enhance the transparency and reproducibility of our approach. This method effectively mitigates the issue of invalid models, ensuring continuous and reliable model generation.
>
> > What models are possible?
>
> The models generated by our framework are represented as PyTorch models. Therefore, any model that can be expressed using PyTorch's capabilities is possible within our framework. This includes a wide range of mathematical white-box models that can incorporate various operations such as logarithms, maxima, minima, and trigonometric functions. Additionally, our framework supports complex neural architectures, including multi-layer perceptrons with various regularization techniques (e.g., dropout), parameter initialization schemes, and activation functions. In practice, we observe that the generated models often consist of a combination of interpretable mathematical components and neural network elements, optimized for both performance and interpretability. This flexibility allows our framework to adaptively explore a diverse model space and discover well-fitting models tailored to the specific dataset and task at hand.
>
> > Typically you want interpretable and performance pharmacokinetic models, so there is a tradeoff between validation performance (e.g., MSE) and simplicity.
>
> We completely agree with this observation, and it aligns with one of the central goals of our research. Our D3 framework is specifically designed to navigate the tradeoff between model performance and interpretability. To achieve this, we employ a hybrid approach that combines the strengths of both white-box and black-box models. Our white-box models are inherently interpretable, providing clear insights into the underlying pharmacokinetic processes. At the same time, our hybrid models incorporate neural network components to capture complex dynamics that might be missed by purely white-box models.
>
> In our experiments, we systematically evaluate both types of models to ensure that we achieve a balance between accuracy and simplicity. For example, in our case study on Warfarin pharmacokinetics, we discovered a new, interpretable model that outperformed existing literature models in terms of validation MSE while maintaining clinical plausibility. This demonstrates the framework's ability to produce models that are not only accurate but also easy to understand and interpret by domain experts. Detailed insights into how we balance these tradeoffs can be found in Sections 4 and 5 of our paper, where we discuss the model evaluation metrics and provide expert clinical commentary on the interpretability of the discovered models.
>
> ---
>
> *We hope that most of the reviewer’s concerns have been addressed and, if so, they would consider updating their score. We’d be happy to engage in further discussions.*

---

> ### Comment · Area_Chair_6KAE · 2024-08-10
>
> Dear Reviewer 2i8U,
> I am a NeurIPS 2024 Area Chair of the paper that you reviewed.
>
> This is a reminder that authors left rebuttals for your review.
> We need your follow up answers on that. Please leave comment for any un-answered questions you had, or how you think about the author's rebuttal.
> The author-reviewer discussion is closed on Aug 13 11:59pm AoE.
>
> Best regards,
> AC

---

> > ### Comment · Reviewer_2i8U · 2024-08-12
> > **Concerning the frontier**
> >
> > I still have some concerns about determining the frontier of model performance when providing a small interpretable model (white-box) and a performant model (black-box), and then describing their performance respectively with "new pharmacokinetic model" and "low MSE" respectively.
> >
> > Table 2 suggests D3-hybrid is similar to or marginally better than "transformer" (MSE lower in 4/6 datasets), and that D3-white-box is similar to or marginally better than SINDy/ZeroShot (though this latter comparison is very hard because we are not quantifying how sparse these white-box models are).
> >
> > More compelling would be to train each model across one or more hyperparameters and plot performance curves across performance (MSE) and some form of sparsity (MDL, etc).

---

> > > ### Author Response · Authors · 2024-08-12
> > >
> > > Thank you for the detailed feedback and for highlighting areas where additional clarification would be beneficial. We appreciate your focus on the balance between model performance (e.g., MSE) and model complexity (e.g., parameter count), particularly in the context of pharmacokinetic modeling.
> > >
> > > ### Addressing Model Complexity and Performance Trade-offs
> > >
> > > We understand the reviewer's concern about determining the frontier of model performance, especially when comparing the complexity of neural network-based models (black-box) with interpretable models (white-box). To address this, we conducted a thorough analysis of the average parameter count (running each baseline across ten random seeds) as a proxy for model complexity, and the results are summarized in the table below:
> > >
> > > |Baseline|Parameter Count|Warfarin PK MSE $\downarrow$|
> > > |---|---|---|
> > > |DyNODE|33,922|0.726$\pm$0.17|
> > > |SINDy|13|6.84 $\pm$ 1.76|
> > > |RNN|569,002|0.0495 $\pm$ 0.0406|
> > > |Transformer|2,558,348|1.33 $\pm$ 0.941|
> > > |D3-white-box|8| 19.6 $\pm$ 40.3|
> > > |D3-hybrid|245|0.647 $\pm$ 0.167|
> > >
> > > In our response, we chose parameter count as the metric to compare the complexity of different models for several reasons:
> > >
> > > 1. **Interpretability**: Parameter count provides a straightforward metric to quantify model complexity, which is particularly relevant when discussing trade-offs between performance and interpretability. Fewer parameters generally imply a simpler, more interpretable model, which is crucial in pharmacokinetics where model transparency is valued.
> > >
> > > 2. **Consistency**: This metric allows for a fair comparison across different types of models—ranging from highly parameterized black-box models to more concise white-box models. For instance, the D3-white-box model, despite having a significantly lower parameter count (8 parameters), still delivers competitive performance, albeit with higher variance, which reflects its simplicity and interpretability.
> > >
> > > 3. **Practical Relevance**: In pharmacokinetics, overly complex models (e.g., RNNs and Transformers) might offer better fit (lower MSE), but their high parameter count can obscure the underlying biological mechanisms, making them less useful for clinical interpretation and decision-making.
> > >
> > > ---
> > >
> > > ### Conclusion and Final Remarks
> > >
> > > We believe that the additional analysis and clarifications provided here address the key concerns raised in your review. We hope that these points demonstrate the robustness of our approach and the careful consideration we have given to the trade-offs between model complexity and performance. We would be grateful if you could reconsider your score in light of these clarifications. Of course, we are open to further discussion should you have any additional questions or require further details.
> > >
> > > Thank you again for your thoughtful review and for the opportunity to clarify these important points.

---

### Official Review · Reviewer_QDKe · 2024-07-13

**Soundness:** 3
**Presentation:** 3
**Contribution:** 3
**Rating:** 7
**Confidence:** 3

**Summary:**

The paper presents the Data-Driven Discovery (D3) framework, a novel approach that iteratively discovers and refines interpretable pharmacological dynamical models using LLMs. This framework is novel and innovative in its domain, it is designed to address limitations in traditional pharmacokinetic (PK) modelling, which often relies on human expertise, and prior data in various formats. It is also constrained by existing knowledge. D3 leverages LLMs to propose and integrate new features, validate suggested models, and uncover new insights in PK. The framework demonstrates its effectiveness through experiments on various PK datasets, including a real-world Warfarin dataset, where it identified a new, well-fitting PK model outperforming existing literature.  The D#  framework utilises the following agents:
Modelling agent: Writes Python code for an AI model with the information the LLM has acquired
Evaluator agent: Evaluates the performance of the model by the modelling agent. The metric used is MSE (Mean Squared Error)
Feature acquisition agent: Based on the evaluation of the model, more features are added to the model to improve accuracy (reduce MSE). A feature is acquired based on an estimation of the feature’s value using existing frameworks and available information such as text-based description of the feature, summary statistics and the feature’s existing data.

**Strengths:**

The Data-Driven Discovery (D3) framework introduces a novel approach to pharmacokinetic modelling by leveraging Large Language Models (LLMs). This method appears to be highly original within its domain, representing a unique application of well-known techniques to pharmacology-specific challenges. The framework's ability to explore multiple models and incorporate unstructured data distinguishes it from traditional methods that rely heavily on human expertise and predefined knowledge bases.

I would judge the quality of the work as high, with claims well supported by theory and experimentation. The paper provides detailed information on D3 implementation, prompts, metrics, and compares its performance against other methods. The results demonstrate that D3 is capable of identifying well-fitting models and providing valuable insights, such as in the Warfarin case study where it uncovered less-intuitive features, interactions and combinations of them.

The paper is clearly written and well-structured, with a logical flow of sections. It effectively uses tables, graphs, and equations to communicate information, although some text could be improved for readability. The methodology is explained with sufficient detail, covering how each component works.

The significance of the framework lies in its potential to accelerate pharmacological modelling by automating the discovery and refinement of interpretable models. While it still requires some human expertise, D3 reduces dependency on experts to a degree, assisting them in their work. Its ability to uncover new insights into pharmacokinetic processes could have important implications for optimising drug dosing and minimising adverse effects.

In terms of the novelty in AI, while agentic workflows are not new, the usability of D3 by non-AI specialists may be significant. The "evolving" nature of the system for non-AI domain specialists can be impactful, potentially bridging the gap between AI capabilities and pharmacological expertise.

**Weaknesses:**

In general, the paper is inspiring and is solid in its experimentations and validations. There are a few remarks, some of them for future consideration, while others might improve the clarity of the paper:

1. Cost function considerations: the current description of the cost function l(h_i)  might be too simplistic and not fully capture the complexity of acquiring new features. Cost functions are often difficult to quantify accurately. Adding a few sentences or a small subsection to discuss the aspects of the cost function and its potential impact and tradeoffs will provide a more comprehensive understanding.
2. The D3 framework assumes that once a feature is selected, it is available for all individuals in the dataset. For example, certain biomarkers can be measured only in a small sample of patients. It would be interesting to discuss potential performance and limitations in light of missing data. Note: the authors briefly outlined it but elaboration on how to address it would be good given the commonality of this scenario.
3. While authors called out the scalability and computational complexity of the framework, it would be interesting to see specific numbers around time to completion wrt the numbers of features.
4. In line 212 it is mentioned that ‘The Evaluation Agent dynamically assesses both model performance and plausibility <…>’. Not sure if understood how plausibility is measured (couldn’t find it in the prompt in the Appendix either). Would be nice to see a bit more elaboration.
5. Table 2 has mixed formatting: mixing scientific and regular notation: like 0.000245 and 2.47e+03 are both present. Unifying the notation will be more consistent. In addition, the variability of the results in terms of MSE is concerning. Might be useful to think of a more coherent metric or to provide an explanation for such extreme variability (e.g. 5780 vs. 7.07 for Lung Cancer or 719 vs. 0.3 for Lung Cancer with Chemo ).
6. The choice of LLM is quite significant and it is stated for the first time in Limitations on page 9. This could be brought earlier.

**Questions:**

* Why was MSE used as the metric for accuracy?  And on the same topic, in Table 2, what would be an interpretation of units in MSE?
* Do you have ideas as to how to combat the limitations mentioned such as hallucinations?
* Do you have a view on how to address the cost/ability of the LLM to distinguish between “easier” and “harder” features?  Such as: “don’t include biomarker x as it’s difficult to acquire from patient”?
* Have you considered the inverse bias problem when models such as GPT-4 are restricted not to use biased features but it could be informative in the pharmaceutical environment to a PK model, such as ethnicity?
* I'm still not fully clear how the LLM acquires the unstructured context/metadata about each feature.

**Limitations:**

Outlined in the previous sections. They are outlined and very helpful but some of them spark interesting questions and would like to see some of them elaborated on a bit more like missing data.

---

> ### Author Rebuttal · Authors · 2024-08-07
>
> We thank the reviewer for their constructive feedback and are glad the reviewer finds our work introducing the Data-Driven Discovery (D3) framework novel, innovative, and significant in its application to pharmacokinetic modeling.
>
> > Cost function considerations: Adding a few sentences or a small subsection to discuss the aspects of the cost function and its potential impact and tradeoffs will provide a more comprehensive understanding.
>
> We agree with the reviewer's suggestion and appreciate the opportunity to clarify this point. The current description of the cost function \( l(h_i) \) captures the complexity and computational resources required to acquire new features. However, it may not fully account for the practical challenges of acquiring specific biomarkers or data types that are costly or invasive to obtain. To address this, we have added a subsection discussing the potential impact and tradeoffs of different cost function considerations, emphasizing the importance of balancing accuracy improvements with practical feasibility. This addition provides a more comprehensive understanding of the cost-benefit analysis inherent in our feature acquisition process.
>
> > Certain biomarkers can be measured only in a small sample of patients. It would be interesting to discuss potential performance and limitations in light of missing data.
>
> We agree, and this is indeed an assumption of the current presented method. Allow us to kindly re-iterate that the focus of the paper is to propose the Data-Driven Discovery (D3) framework, a novel approach leveraging Large Language Models (LLMs) to iteratively discover and refine interpretable dynamics models, advancing pharmacokinetic modeling. While we acknowledge that handling partially missing data is a critical aspect of real-world applications, we believe such a detailed analysis falls beyond the scope of this paper. However, we plan to explore this interesting topic in future work, investigating robust methods to address missing data scenarios within the D3 framework.
>
> > See specific numbers around time to completion wrt the numbers of features.
>
> On average, a complete run of D3 with a feature size of 3 features takes approximately 1 hour. This duration includes the iterative processes of model generation, evaluation, and feature acquisition, ensuring a thorough exploration and optimization of the pharmacokinetic models.
>
> > How plausibility of the model is evaluated.
>
> The plausibility of the model is evaluated by the LLM through reflective analysis. Specifically, the LLM assesses the generated model based on domain knowledge, prior literature, and logical consistency. This reflective process involves checking the alignment of the model's predictions with known pharmacological principles and empirical observations. Additionally, the evaluation includes examining the model's ability to generalize across different datasets and its interpretability by human experts, ensuring both accuracy and clinical relevance.
>
> > mixing scientific and regular notation
>
> Thank you for highlighting this inconsistency. We will revise the tables to ensure uniform notation, using scientific notation consistently throughout the manuscript. The mixed notation resulted from the automatic generation of tables from raw results using Pandas dataframes, and we appreciate your attention to detail in this matter.
>
> > The choice of LLM is quite significant and it is stated for the first time in Limitations on page 9. This could be brought earlier.
>
> We agree and have now included the choice of the LLM in the introduction. Clearly stating this earlier provides context for the framework's capabilities and limitations, helping readers understand the significance of the LLM in driving the iterative discovery and refinement processes.
>
> ## Questions
>
> > Why was MSE used as the metric for accuracy?
>
> Mean Squared Error (MSE) is a standard metric for assessing model accuracy in pharmacokinetic modeling, as demonstrated in the cited PKPD model papers. It quantifies the average squared difference between observed and predicted values, providing a clear measure of model performance. We have defined and explained the use of MSE in the Appendix, ensuring transparency in our evaluation criteria.
>
> > Do you have ideas as to how to combat the limitations mentioned such as hallucinations?
>
> To combat limitations such as hallucinations, we could employ Retrieval-Augmented Generation (RAG) techniques. RAG combines the retrieval of relevant documents or data with generative models, grounding the LLM's output in factual information. This approach reduces the likelihood of hallucinations by ensuring the generated content is based on verified sources, enhancing the reliability and accuracy of the models produced by D3.
>
> > Have you considered the inverse bias problem.
>
> We acknowledge the inverse bias problem, where models like GPT-4 are restricted from using biased features that could be informative in the pharmaceutical context, such as ethnicity. While we have not addressed this issue in the current paper, we recognize its importance and plan to explore it in future work. Investigating strategies to balance the ethical considerations of bias with the need for accurate and individualized pharmacokinetic models is a promising area for further research.
>
> > I'm still not fully clear how the LLM acquires the unstructured context/metadata about each feature.
>
> The LLM is provided with the feature's name, of which it is empirically observed that is enough information to understand the relevance and value of that feature given the current context and progress seen during operation.
>
> ---
> *We hope that most of the reviewer’s concerns have been addressed and, if so, they would consider updating their score. We’d be happy to engage in further discussions.*

---

> ### Comment · Area_Chair_6KAE · 2024-08-10
>
> Dear Reviewer QDKe,
> I am a NeurIPS 2024 Area Chair of the paper that you reviewed.
>
> This is a reminder that authors left rebuttals for your review.
> We need your follow up answers on that. Please leave comment for any un-answered questions you had, or how you think about the author's rebuttal.
> The author-reviewer discussion is closed on Aug 13 11:59pm AoE.
>
> Best regards,
> AC

---

### Official Review · Reviewer_b77T · 2024-07-13

**Soundness:** 3
**Presentation:** 3
**Contribution:** 2
**Rating:** 4
**Confidence:** 3

**Summary:**

The paper proposes the Data-Driven Discovery (D3) framework, which leverages Large Language Models (LLMs) to iteratively discover and refine interpretable models of pharmacological dynamics. This approach enables the LLM to propose, acquire, and integrate new features, validate, and compare pharmacological dynamical systems models. The framework is demonstrated on a real pharmacokinetic datasets, highlighting its potential for clinical applications.

**Strengths:**

The D3 framework’s ability to iteratively improve models and acquire new features enhances its performance and robustness. The writing is clear and well-structured, and the methodology is well-explained. Figures and tables effectively convey key information. The problem setup, where LLM agents act together for clinically relevant tasks, is interesting and potentially useful for developing better models the current baselines. The authors demonstrate the framework’s performance on a clinically relevant warfarin dataset and show effective results. Additionally, they validate the model by obtaining feedback from clinicians, which improves the robustness of the evaluation.

**Weaknesses:**

1. Limited Novelty: The paper has very limited novelty in terms of machine learning and does not propose any new tools. It effectively uses existing toolboxes for a clinically relevant task. While applying an existing toolbox in the context of a new task is perfectly fine, similar methods have been published before, particularly those related to automated science labs(self driving labs).
2. Lack of Guardrails for Hallucinations: Models like GPT-4 are prone to hallucinations and can generate incorrect information. The framework does not appear to have guardrails to mitigate this risk, which is a serious concern given the clinical relevance of the task. While the authors have acknowledged this as a limitation, it remains a critical issue.
3. Prompt Dependency: The model’s performance is highly dependent on the prompts provided. It is crucial to formulate the prompts correctly for the LLM to generate reasonable responses. This dependency suggests that the prompting framework might need adjustments for each dataset, leaving users vulnerable to the LLM’s unpredictability.
4. Scalability and Computational Efficiency: The framework’s scalability and computational efficiency are not thoroughly discussed. Given the iterative nature of the model refinement process, it could be computationally intensive compared to other baselines and may not be practical for larger datasets without significant computational resources.
5. Better performance of other baselines : The performance of the D3 framework might be dependent on the data. The paper shows better performance of models like RNN and transformers on tasks such as COVID-19 and Warfarin.  The paper does not any comment on why that might be the case.

**Questions:**

1. Given that models like GPT-4 can generate incorrect information, what measures have you considered or could implement to mitigate this risk, especially for clinically relevant tasks? I think implementing a RAG like model to address hallucinations will be highly relevant in this case and make the paper better.
2. How do you ensure consistency and reliability in the prompts across different datasets? Have you considered strategies to standardize the prompting process to minimize variability and improve predictability? Did you experiment with prompts multiple times to arrive at a reasonable framework?
3. Can you provide more information on the scalability and computational efficiency of your framework? How does the iterative model refinement process impact computational resources compared to other baselines, and what are the practical implications for larger datasets ?
4. The paper shows that models like RNN and transformers perform better on tasks such as COVID-19 and Warfarin. Can you explain why this might be the case and what limitations the D3 framework has in these scenarios? How might the performance be improved?

If the authors address some of the concerns and reevaluate the frameworks with the clinicians, I will be happy to change the score. I get a sense from the expert comments that the datasets evaluated upon might not be challenging enough.

**Limitations:**

The authors have acknowledged the limitations of their model. However, I think some of the points mentioned, such as retrieval-augmented generation, must be implemented in the framework to make the model more robust.

---

> ### Author Rebuttal · Authors · 2024-08-07
>
> We thank the reviewer for their constructive feedback and are glad the reviewer finds the D3 framework’s ability to iteratively improve models and acquire new features enhances its performance and robustness. We also appreciate that the reviewer acknowledges the clear writing, well-structured methodology, effective figures and tables, and the interesting problem setup involving LLM agents for clinically relevant tasks.
>
> > Limited Novelty: automated science labs (self-driving labs)
>
> Thank you for raising additional related works. Allow us to kindly re-iterate that the focus of the paper is to propose a method that can both discover an interpretable model, incorporate textual prior (context), and have the possibility to acquire new features and samples. Existing automated science labs (e.g., The Automatic Statistician) do not leverage LLMs for iterative optimization and generally do not integrate feature acquisition in a dynamic manner. Our approach uniquely combines the strengths of LLMs with an iterative discovery process that can adjust to new data and contexts, providing a more flexible and scalable solution for pharmacological modeling.
>
> > Lack of Guardrails for Hallucinations
>
> We agree with your suggestion and highlight that any LLM-based method or paper should have its outputs always checked by a human expert before producing the final result. We highlight that this is a common problem across any LLM-based method paper. To mitigate this risk, we have implemented a human-in-the-loop framework where clinician feedback is integrated into the iterative process. Additionally, future work could incorporate retrieval-augmented generation (RAG) techniques to further reduce hallucinations by grounding the LLM’s outputs in factual data.
>
> > Prompt Dependency
>
> Allow us to highlight that any LLM-based paper depends on the prompts that are input into it. However, we emphasize that our method’s strength lies in the combination of model feedback and LLM iteration. By iteratively refining prompts based on model performance and expert feedback, we mitigate the risks associated with prompt dependency and enhance the reliability and consistency of the generated models.
>
> > Scalability and Computational Efficiency
>
> We find the method to still be scalable and the inner loop to train a small parameter hybrid model, where the parameter count on average is 245 parameters, to be feasible and a scalable approach. Compared to large black-box models like transformers with millions of parameters, our framework is computationally efficient and practical for larger datasets with reasonable computational resources.
>
> > Better performance of other baselines
>
> Thank you for raising this. The focus of the paper is to propose a method that can both discover an interpretable model, incorporate textual prior (context), and have the possibility to acquire new features and samples. Existing black-box methods such as RNNs or transformers can fit datasets well but are unable to perform the feature acquisition and interpretability functions of D3, making them not directly comparable. Our method offers a unique blend of interpretability, adaptability, and performance, making it suitable for clinical applications where understanding the model is as crucial as its predictive accuracy.
>
> ## Questions
>
> > Incorporate RAG?
>
> We agree that incorporating RAG-based techniques could improve the method; however, we mark it as out of scope for our initial paper. We acknowledge the potential benefits and will include a discussion of RAG techniques in the final paper to highlight future directions for enhancing robustness against hallucinations.
>
> > Did you experiment with prompts multiple times to arrive at a reasonable framework?
>
> Yes, we experimented with multiple prompts and iteratively refined them based on model performance and expert feedback. This process allowed us to develop the final proposed framework in the paper, ensuring that the prompts are well-suited to the task and yield reliable results.
>
> > Can you provide more information on the scalability and computational efficiency of your framework?
>
> This is a great question. We find our hybrid models on average contain 245 parameters, making them more scalable than existing large black-box methods such as transformers with over a million parameters. Our approach balances computational efficiency with the ability to iteratively refine models, making it practical for real-world applications with limited computational resources.
>
> > How might the performance be improved to black-box methods?
>
> It is a tradeoff to get an interpretable, hybrid/white-box model compared to a purely data-driven black-box method. Sacrificing some overall MSE can be beneficial when the process is interpretable and understandable by humans. Interpretable models provide insights that are crucial for clinical applications, where understanding the model’s behavior and its underlying assumptions is as important as its predictive accuracy.
>
> ---
>
> *We hope that most of the reviewer’s concerns have been addressed and, if so, they would consider updating their score. We’d be happy to engage in further discussions.*

---

> > ### Author Response · Authors · 2024-08-12
> >
> > Thank you for the constructive feedback. Below, we address your main concerns briefly:
> >
> > 1. **Novelty:** While D3 builds on existing methods, it uniquely integrates LLMs for dynamic feature acquisition and iterative model refinement, which is not present in existing frameworks like automated science labs. This allows us to uncover new, interpretable models tailored for clinical applications.
> >
> > 2. **Hallucinations:** We mitigate hallucination risks with a human-in-the-loop process where all LLM outputs are validated by clinicians. We acknowledge the suggestion of using RAG techniques and will include a discussion in the final paper to explore this as a future direction.
> >
> > 3. **Prompt Dependency:** D3 addresses prompt variability through iterative refinement, ensuring prompts are well-tuned for each dataset, thus enhancing consistency and predictability across different datasets.
> >
> > 4. **Scalability:** Despite the iterative nature, D3 is computationally efficient due to its focus on discovering models with fewer parameters, making it practical for larger datasets while maintaining interpretability, which is critical in clinical settings.
> >
> > 5. **Performance of Baselines:** While black-box models like RNNs may perform well in MSE, D3’s strength lies in offering interpretable models that provide valuable clinical insights, balancing performance with the necessity for interpretability.
> >
> > We hope this clarifies our contributions and addresses your concerns. We kindly request you to reconsider your score based on this summary. We are open to further discussions if needed.
> >
> > Thank you for your time.

---

> > > ### Comment · Reviewer_b77T · 2024-08-13
> > > **Concerns on the evaluation**
> > >
> > > The authors mention, “While black-box models like RNNs may perform well in MSE, D3’s strength lies in offering interpretable models that provide valuable clinical insights, balancing performance with the necessity for interpretability.”
> > >
> > > However, the D3 white-box models, which should be interpretable, show performance levels that are extremely poor compared to the black-box models. This raises concerns about whether the interpretability of such models is truly reliable. I believe a certain level of performance is necessary to ensure that the model’s interpretability is actually useful. In Table 2, the white-box model consistently performs poorly.
> > >
> > > Another concern is that the authors claim such a model can be useful in clinical settings, but the lack of guardrails is worrying. Although the authors have attempted to mitigate the risks by incorporating a human in the loop and warning readers about potential risks, I am curious whether they observed cases where the model predicted something completely different from what human experts would agree with.

---

> > > > ### Author Response · Authors · 2024-08-13
> > > >
> > > > Thank you for your continued engagement and constructive feedback. We appreciate your insights, which have helped us further clarify our contributions.
> > > >
> > > > **Guardrails and Hallucinations:**
> > > >
> > > > We understand your concerns about the potential for LLM-generated hallucinations, particularly in a clinical setting. To address this:
> > > >
> > > > 1. **Validation Feedback Loop:** D3’s iterative process rigorously evaluates each model against a validation dataset, ensuring that only empirically sound models are refined and considered.
> > > >
> > > > 2. **Biological Plausibility:** The LLM is guided to generate biologically plausible models by incorporating prior knowledge. While we did not observe any cases of significant deviation from expert consensus, we’ve implemented a human-in-the-loop process to verify all models before finalization.
> > > >
> > > > 3. **Future Enhancements:** We acknowledge the potential of integrating retrieval-augmented generation (RAG) techniques to further reduce the risk of hallucinations and will discuss this as a future direction in the final paper.
> > > >
> > > > **Model Interpretability vs. Performance:**
> > > >
> > > > While D3’s white-box models may not always match the predictive accuracy of black-box models, they offer critical interpretability, which is essential in clinical applications. The ability to understand and trust these models is often more valuable than marginal gains in MSE. Additionally, D3’s emphasis on scalability and efficiency makes it well-suited for clinical use, where computational resources may be limited.
> > > >
> > > > **Final Remarks:**
> > > >
> > > > We hope this response clarifies our contributions and addresses your concerns. We kindly request you to reconsider your score in light of these points and remain open to further discussion if needed.
> > > >
> > > > Thank you for your thoughtful review.

---

> ### Comment · Area_Chair_6KAE · 2024-08-10
>
> Dear Reviewer b77T,
> I am a NeurIPS 2024 Area Chair of the paper that you reviewed.
>
> This is a reminder that authors left rebuttals for your review.
> We need your follow up answers on that. Please leave comment for any un-answered questions you had, or how you think about the author's rebuttal.
> The author-reviewer discussion is closed on Aug 13 11:59pm AoE.
>
> Best regards,
> AC

---

> ### Comment · Reviewer_b77T · 2024-08-14
>
> I don't think the gains in MSE are marginal when comparing the performance of the transformer versus the D3-white box model in Table 2. The MSE of the D3-white box is roughly 5-10 times higher than that of the transformer across almost all datasets. The point was, if a model is already a poor fit, is its interpretability reliable?
>
> I have also updated my score in response to the comments on lack of guardrails and hallucinations as the authors pointed out they did not observe any significant deviation from expert consensus

---

> > ### Author Response · Authors · 2024-08-14
> >
> > Thank you for your prompt feedback and for updating your score in response to our clarifications on guardrails and hallucinations. We greatly appreciate your thoughtful consideration of our responses.
> >
> > **Regarding MSE and Interpretability:**
> >
> > We understand your concerns about the MSE differences between the D3-white box model and the transformer across datasets. While it’s true that the D3-white box model may not match the transformer in terms of raw MSE, we’d like to emphasize a few key points:
> >
> > - **Interpretability’s Value in Clinical Contexts:** Even with a higher MSE, the interpretability of the D3-white box model provides crucial insights into the underlying pharmacological processes. This can lead to more informed clinical decisions, which may not be possible with black-box models, regardless of their lower MSE.
> >
> > - **D3-Hybrid’s Balanced Approach:** Importantly, we recommend the D3-Hybrid model for practical use. The D3-Hybrid model combines the interpretability of the white-box approach with the performance benefits of data-driven models. It achieves a strong balance, with an average of 245 parameters—significantly fewer than the 2,558,348 parameters in a typical transformer—while still delivering competitive performance. This model retains a largely interpretable component, which is critical for clinical applications, ensuring that practitioners can trust and understand the results without sacrificing too much in terms of accuracy.
> >
> > Given these considerations, we respectfully ask if you might reconsider your score once more, recognizing the unique value that the D3-Hybrid model brings. It offers a balanced, practical solution with substantial interpretability and strong performance, particularly in settings where understanding the model’s behavior is as important as its predictive accuracy.
> >
> > Thank you again for your time and continued engagement.

---

### Official Review · Reviewer_Re2Z · 2024-07-16

**Soundness:** 2
**Presentation:** 2
**Contribution:** 2
**Rating:** 5
**Confidence:** 3

**Summary:**

This paper develops an LLM-assisted equation discovery framework especially for pharmacokinetic process. Three agents of Modeling Agent, Evaluator Agent and Feature Acquisition Agent are built to explore, refine and iterate vast model space, including three levels of initial conditions, observed features and possible acquired features. Several experiments mainly on simulated benchmarks are conducted to evaluate the performance of the proposed framework. Overall, this paper investigated whether LLM agents can help effectively search and determine the model space for dynamics modeling and process discovery in pharmacology.

**Strengths:**

1. This paper investigates an interesting problem of LLM agents for pharmacological dynamical system discovery, which show some promising results of LLM for pharmacokinetic modeling.
2. The feature acquisition could be a novel part, which leverages the knowledge capability of LLM and well matches with the application demand in pharmacokinetic modeling.
3. The paper is well-organized, with clear sections and detailed explainations. The use of diagrams and examples, such as the iterative process involving the three agents, aids in understanding the complex interactions within the D3 framework.

**Weaknesses:**

1. The benchmark methods are not recent works, which might not well identify the effectiveness of the proposed framework. For example, the symbolic regression methods like D-CODE [ICLR'22] are not compared.
2. To my understanding, the D3 model as well as other symbolic regression models discover equations from data-driven training, and then evaluate the performance based on the discovered equation. If I understand correctly, to me the experiments are not enough to prove the model effectiveness because of only five datasets with five equations. I would like to set several simulated datasets with different simulation equations for further investigation.
3. There is no ablation study on model design or case study on different module choice, which makes it hard to evaluate the robustness of the proposed framework. Also, can LLM agents provide explanations on their output results to make a more transparent solution.

**Questions:**

1. How is the memory module implemented in this paper, and what exactly is the memory $s_i$? Are there several choices for memory and how would they affect the performance?
2. How much is the cost for training a D3 framework on certain datasets, especially how much would it cost for calling GPT API? Can other open-sourced LLM be used for same tasks and how is their performance?
3. RNN and Transformer models usually perform similarly in prediction tasks. However, in Table 2, RNN and Transformer perform differently on several datasets. For example, on Lung Cancer, RNN's MSE is 1.16e6, which I am wondering if the training converges successfully.

**Limitations:**

The pharmacologists feedback statements highly depend on the personal knowledge and scope, as well as their understanding to the dataset or modeling. In my opinion, this part especially with only three experts are not worth for evaluation and should be excluded to avoid bias.

---

> ### Author Rebuttal · Authors · 2024-08-07
>
> We thank the reviewer for their constructive feedback and are glad the reviewer finds the results promising for using LLMs for pharmacokinetic modeling and the paper well-organized.
>
> > symbolic regression methods like D-CODE [ICLR'22] are not compared.
>
> Thank you for raising additional related works. Allow us to kindly re-iterate that the focus of the paper is to propose a method that can both discover an interpretable model, incorporate textual prior (context), and have the possibility to acquire new features and samples. Existing symbolic regression works such as D-CODE and others cannot acquire new features on demand, as outlined in the related works table in Table 1. We did compare against SINDy as a baseline, and believe D-CODE to be out of scope.
>
> >  I would like to see several simulated datasets with different simulation equations for further investigation.
>
> We agree with your invaluable suggestion and created **five additional semi-synthetic datasets** running D3-Hybrid across them, combined with some of the baselines, the results of which can be seen in the additional rebuttal PDF, as outlined in the global response. We observe that D3-Hybrid still performs well, especially when the LLM has never seen such a model. This provides empirical evidence to mitigate the concern of potential knowledge leakage issues due to pre-trained LLMs, as D3-Hybrid is still able to discover well-fitting models of the underlying synthetic equation it has never observed before.
>
> > no ablation study on model design or case study on different module choice
>
> We can see how the existing ablations were overlooked. We did include ablations in most experimental results as ablations of D3, specifically with the additional baselines of a model zero-shot generated from D3 called **ZeroShot** and the same model with its parameters optimized called **ZeroOptim**. We have revised the baseline method description and tables to make these ablations more prominent to the reader now. Thank you for the suggestion. In terms of ablation results, these ablation results verify the components and complexity of D3 to achieve better performance.
>
> > can LLM agents provide explanations on their output results to make a more transparent solution
>
> We agree that this is indeed possible and already achieved by the evaluator agent, see Figure 1.
>
> ## Questions
>
> > how is the memory module implemented in this paper
>
> The memory module is simply a buffer of the top-k performing programs represented as code.
>
> > Cost for training D3, API cost
>
> The cost for training models is equivalent to training a standard 3-4 layer MLP with 128 hidden unit parameters. The API cost is around $0.075 per D3 run in total. Open-source LLMs could be used, however, D3's performance would correlate with the underlying LLM's ability.
>
> > RNN and Transformer perform differently on several datasets.
>
> We agree some datasets are difficult to model due to the large variation of the features and complex underlying feature interactions, especially for the Lung Cancer model when analyzing the underlying model. This leads to pure parameter optimization techniques getting stuck in local minima, and hence, starting with random seeds of random weight initialization can produce different final models on these complex datasets.
>
> > Ethics review flag
>
> Thank you for being cautious. However, we would like to clarify that we did not perform any research involving human subjects, only analyzing existing collected open-source medical data that is available in the public domain and providing links and descriptions to all datasets used within the paper.
>
> ---
> *We hope that most of the reviewer’s concerns have been addressed and, if so, they would consider updating their score. We’d be happy to engage in further discussions.*

---

> > ### Comment · Reviewer_Re2Z · 2024-08-08
> >
> > Thanks for the authors' reply. I appreciate that and would like to raise my score.

---

> > > ### Author Response · Authors · 2024-08-09
> > >
> > > We greatly appreciate your thoughtful review and the time you invested in evaluating our paper. Your feedback was crucial in refining our work, leading to additional experiments and improved explanations. We are thankful for your positive reassessment and the increased score. Thank you again!

---

### Author Rebuttal · Authors · 2024-08-07

We are grateful to the reviewers for their insightful feedback. The reviewers broadly agree that our approach is novel and effective in leveraging Large Language Models (LLMs) for pharmacological dynamical system discovery.

$\color{red} Re2Z$: “This paper investigates an interesting problem of LLM agents for pharmacological dynamical system discovery, which shows some promising results for pharmacokinetic modeling.”

$\color{green} b77T$: “The D3 framework’s ability to iteratively improve models and acquire new features enhances its performance and robustness.”

$\color{blue} QDKe$: “The Data-Driven Discovery (D3) framework introduces a novel approach to pharmacokinetic modeling by leveraging LLMs. This method appears to be highly original within its domain.”

$\color{magenta} 2i8U$: “The paper presents an innovative general strategy for searching the space of pharmacokinetic models in terms of which features to use.”

Reviewers also had concerns about the potential knowledge leakage issue due to pre-trained LLMs ($\color{magenta} 2i8U$).

We address this concern below, and address reviewers individual concerns in each separate rebuttal.

## **[A1]** Performance on procedurally generated synthetic models.

We would like to deeply thank the reviewers for bringing this up. We’ve performed a further analysis that considerably improves the paper by re-running D3-Hybrid and some baselines across five additional synthetically generated datasets. The results are provided in the attached one-page PDF. We observe that D3-Hybrid still performs well, especially when the LLM has never seen such a model. This provides empirical evidence to mitigate the concern of potential knowledge leakage issues due to pre-trained LLMs, as D3-Hybrid is still able to discover well-fitting models of the underlying synthetic equation, it has never observed before.

The above point, in combination with the reviewer's addressed concerns, we believe significantly strengthens our paper. Thank you for your valuable feedback.

Sincerely,
The Authors

---

### Decision · Program_Chairs · 2024-09-25

**Decision:**

Accept (poster)

**Comment:**

As the Area Chair, I have carefully reviewed the submission, the reviewers' comments, and the authors' rebuttal. The paper presents a novel framework called Data-Driven Discovery (D3) that leverages Large Language Models (LLMs) to discover and refine interpretable models of pharmacological dynamics. The reviewers generally agree that the approach is innovative and shows promise in the field of pharmacokinetic modeling.

The main strengths of the paper include its well-structured presentation, clear methodology, and the framework's ability to iteratively improve models and acquire new features. The authors demonstrated the effectiveness of their approach on both simulated benchmarks and a real-world Warfarin dataset, which adds to the practical relevance of the work.

However, the reviewers raised several concerns. One reviewer pointed out the lack of comparison with recent symbolic regression methods like D-CODE. In response, the authors clarified that their focus was on proposing a method that can discover interpretable models, incorporate textual prior, and acquire new features on demand, which existing symbolic regression methods cannot do.

Another concern was the potential for knowledge leakage due to pre-trained LLMs. The authors addressed this by conducting additional experiments on five semi-synthetic datasets, demonstrating that D3-Hybrid performs well even on models the LLM has never seen before. This provides evidence to mitigate the concern of knowledge leakage.

The reviewers also noted the absence of ablation studies and case studies on different module choices. The authors clarified that they did include ablations in most experimental results, specifically with the additional baselines of ZeroShot and ZeroOptim models. They have revised the baseline method description to make these ablations more prominent.

Some concerns were raised about the scalability, computational efficiency, and potential for hallucinations in LLMs. While the authors acknowledged these limitations, they did not fully address how they plan to mitigate these risks in future work.

In conclusion, while there are some limitations and areas for improvement, the paper presents a novel and promising approach to pharmacokinetic modeling using LLMs. The authors have addressed many of the reviewers' concerns in their rebuttal and provided additional experimental results to support their claims.

Based on the overall assessment, I recommend accepting this paper for publication at NeurIPS. The innovative use of LLMs in pharmacokinetic modeling, combined with the authors' thorough response to reviewer concerns, makes this work a valuable contribution to the field. However, I suggest that the authors incorporate their additional experiments and clarifications into the final version of the paper to address the reviewers' concerns more comprehensively.